# East Asian methane emissions inferred from high-resolution inversions of GOSAT and TROPOMI observations: a comparative and evaluative analysis

Ruosi Liang[1,2,3], Yuzhong Zhang[2,3], Wei Chen[1,2,3], Peixuan Zhang[2,3,4], Jingran Liu[1,2,3], Cuihong Chen[5], Huiqin Mao[5], Guofeng Shen[6], Zhen Qu[7], Zichong Chen[8], Minqiang Zhou[9], Pucai Wang[9,10], Robert J. Parker[11,12], Hartmut Boesch[11,12], Alba Lorente[13], Joannes D. Maasakkers[13], Ilse Aben[13]

[1]College of Environmental and Resource Sciences, Zhejiang University, Hangzhou, Zhejiang 310058, China

[2]Key Laboratory of Coastal Environment and Resources of Zhejiang Province, School of Engineering, Westlake University, Hangzhou, Zhejiang 310030, China

[3]Institute of Advanced Technology, Westlake Institute for Advanced Study, Hangzhou, Zhejiang 310024, China

[4]Fudan University, Shanghai 200433, China

[5]Center for Satellite Application on Ecology and Environment, Ministry of Ecology and Environment of China, Beijing 100094, China

[6]Laboratory for Earth Surface Processes, College of Urban and Environmental Sciences, Peking University, Beijing 100871, China

[7]Department of Marine Earth and Atmospheric Sciences, North Carolina State University, Raleigh, NC 27607, USA

[8]School of Engineering and Applied Science, Harvard University, Cambridge, MA, USA

[9]Institute of Atmospheric Physics, Chinese Academy of Sciences, Beijing 100029, China

[10]University of Chinese Academy of Sciences, Beijing 100049, China

[11]National Centre for Earth Observation, University of Leicester, Leicester, UK

[12]Earth Observation Science, School of Physics and Astronomy, University of Leicester, Leicester, UK

[13]SRON Netherlands Institute for Space Research, Utrecht, the Netherlands

*Correspondence to*: Yuzhong Zhang (zhangyuzhong@westlake.edu.cn)

**Abstract.** We apply atmospheric methane column retrievals from two different satellite instruments (GOSAT and TROPOMI) to a regional inversion framework to quantify East Asian methane emissions for 2019 at $0.5° \times 0.625°$ horizontal resolution. The goal is to assess if GOSAT (relatively mature but sparse) and TROPOMI (new and dense) observations inform consistent methane emissions from East Asia with identically configured inversions. Comparison of the results from the two inversions show similar correction patterns to the prior inventory in Central North China, Central South China, Northeast China, and Bangladesh, with less than 2.6 Tg a$^{-1}$ differences in regional posterior emissions. The two inversions, however, disagree over some important regions particularly in northern India and East China. The methane emissions inferred from GOSAT observations are 7.7 Tg a$^{-1}$ higher than those from TROPOMI observations over northern India but 6.4 Tg a$^{-1}$ lower over East China. The discrepancies between the two inversions are robust against varied inversion configurations (i.e., assimilation window and error specifications). We find that the lower methane emissions from East China inferred by the GOSAT inversion are more consistent with independent ground-based *in situ* and total column (TCCON) observations, indicating that the TROPOMI retrievals may have high XCH$_4$ biases in this region. We also evaluate inversion results against tropospheric aircraft

observations over India during 2012–2014 by using a consistent GOSAT inversion of earlier years as an inter-comparison platform. This indirect evaluation favors lower methane emissions from northern India inferred by the TROPOMI inversion. We find that in this case the discrepancy in emission inference is contributed by differences in data coverage (almost no observations by GOSAT vs. good spatial coverage by TROPOMI) over the Indo-Gangetic Plain. The two inversions also differ substantially in their posterior estimates for Northwest China and neighboring Kazakhstan, which is mainly due to seasonally varying biases between GOSAT and TROPOMI $XCH_4$ data that correlate with changes in surface albedo.

## 1 Introduction

Methane ($CH_4$) is a powerful greenhouse gas, with a global warming potential ~80 times that of carbon dioxide ($CO_2$) on a 20-year timescale and ~30 times on a 100-year timescale (Forster et al., 2021). In 2020, the atmospheric methane concentration has increased to 1889 ± 2 ppbv, 262% of pre-industrial levels in 1750, driven primarily by increasing anthropogenic emissions (WMO, 2021). The last decade has seen a rapid growth of atmospheric methane (~8.6 ppbv $a^{-1}$), after a brief period of stabilization in the early 2000s (Dlugokencky et al., 2011; Fletcher and Schaefer, 2019; Rigby et al., 2008; Yin et al., 2021; Zhang et al., 2021). Rising methane concentrations, if continued at current rates in coming decades, may negate benefits of $CO_2$ emission reduction and therefore curbing methane emissions in the 2020s is vital for the success of the Paris Agreement (Ganesan et al., 2019; Nisbet et al., 2019).

Information on methane emissions is required at global, national, and regional levels to guide climate actions on methane. Current bottom-up inventories are often inadequate for this purpose because of their large uncertainties in emission factors and lack of information on emission activities (Saunois et al., 2020). Independent measurements of atmospheric methane, including those from satellite remote sensing, are thus used to evaluate and improve these bottom-up inventories (Jacob et al., 2016). This is generally done through an inversion of atmospheric observations with a chemical transport model to characterize the relationship between emissions and concentrations. Atmospheric methane is measured by two classes of satellite instruments, point source imagers and area flux mappers. While point sources imagers (e.g., Sentinel-2, Landsat, GHGSat) specialize in detecting large emissions from point sources, area flux mappers provide high-precision measurements that can be used to constrain methane fluxes on regional and global scales (Jacob et al., 2022). Area flux mappers that are currently in operation include the TANSO-FTS instrument onboard the Greenhouse gases Observing SATellite (GOSAT) launched in 2009 (Kuze et al., 2016) and the more recent TROPOspheric Monitoring Instrument (TROPOMI) onboard the Sentinel 5 Precursor (S5P) satellite launched in 2017 (Hu et al., 2016; Lorente et al., 2021; Veefkind et al., 2012). Satellite observations made by these area flux mappers are especially valuable in constraining methane emissions over regions with no or only sparse ground networks, including Africa, South America, and East and South Asia (Lu et al., 2021).

Both GOSAT and TROPOMI operate in sun-synchronous orbits and retrieve column-averaged dry-air methane mole fractions ($XCH_4$) from backscattered solar shortwave infrared radiation. TROPOMI continuously images the land surface at a pixel resolution of 7 km × 7 km (5.5 km × 7 km after August 2019) with daily global coverage (Hu et al., 2018; Lorente et al., 2021; Sha et al., 2021), while GOSAT in its standard-viewing mode measures with a 3 day return time in 10 km diameter circular footprints that are typically spaced ~250 km apart (Butz et al., 2011; Kuze et al., 2009; Kuze et al., 2016; Yokota et al., 2009). As a result of differing sampling strategies, TROPOMI generates much higher observation density than GOSAT, which in principle should benefit fine-resolution inversions. The two instruments also measure at different wavelengths, GOSAT at the 1.65 μm band and TROPOMI at the 2.3 μm band. This affects the algorithm that can be applied to retrieve $XCH_4$. Operational TROPOMI retrievals use the RemoTeC full-physics method (Hu et al., 2018). The method is prone to spatially and temporally variable biases owing to scattering artefacts (Hu et al., 2018; Lorente et al., 2021; Sha et al., 2021). These biases in general are not reducible with more observations and, if not corrected, can translate into biases in emission estimates in an inversion. Because of spectrally adjacent $CO_2$ and $CH_4$ absorption in the 1.65 μm band, GOSAT retrievals can alternatively use the $CO_2$ proxy method, in which $XCH_4$ is derived from directly retrieved $CH_4$ to $CO_2$ column ratios and independently specified (simulated or assimilated) $CO_2$ columns (Alexe et al., 2015; Frankenberg et al., 2005; Frankenberg et al., 2006; Parker et al., 2015; Parker et al., 2020). The proxy method usually results in reduced variable biases, as scattering artefacts largely cancel out in retrieving $CH_4$ to $CO_2$ column ratios. It is, however, subject to any errors in specified $CO_2$ columns. The proxy method also leads to a better retrieval success rate over regions with high aerosol loadings or thin clouds, as the method is less sensitive to these interferences compared to the full-physics approach.

A number of studies have applied GOSAT data in inversions on a range of scales (Alexe et al., 2015; Cressot et al., 2014; Feng et al., 2022; Lu et al., 2021; Maasakkers et al., 2019; Monteil et al., 2013; Pandey et al., 2016; Turner et al., 2015; Zhang et al., 2021). TROPOMI data have also been applied in several regional inversion studies (Chen et al., 2022; McNorton et al., 2022; Shen et al., 2021; Shen et al., 2022; Zhang et al., 2020) often with the focus on resolving fine-scale emission hotspots. Qu et al. (2021) performed global inversions of GOSAT and TROPOMI observations at 2° × 2.5° resolution in a comparative analysis, and they showed that methane emissions inferred from the two inversions are generally consistent on the global scale but with significant regional discrepancies including over China.

Here we present high-resolution (0.5° × 0.625°) inversions of GOSAT and TROPOMI observations over East Asia. The main objective is to assess the consistency of methane fluxes inferred from the two sets of satellite data that differ in their data coverage and regional bias (Qu et al., 2021), adding information to the uncertainty characterization of satellite-based methane emission accounting. We perform the analyses with identically configured inversions to isolate the effects of observation data, and we further use independent ground-based observations to evaluate the discrepancies between the two inversions and discuss the cause of differences. This study focuses on East Asia (including China and northern India), which is one of the world's major methane emitting regions and accounts for more than 20% of global emissions (UNFCCC, 2020). The region

has been an important contributor to global increases in methane emissions, but the magnitude of the trend and its sectoral attributions are debated (Ganesan et al., 2017; Gao et al., 2021; Liu et al., 2021; Miller et al., 2019; Sheng et al., 2021; Zhang et al., 2021).

## 2 Observation Data

### 2.1 Satellite observations

We used $XCH_4$ observations from GOSAT and TROPOMI for 2019 in regional inversions over East Asia. For GOSAT, we use the University of Leicester Proxy $XCH_4$ v9.0 retrievals (Parker and Boesch, 2020). Our inversion assimilates only high-quality GOSAT retrievals flagged as "xch4_quality_flag=0" over both land and ocean (glint mode). This GOSAT product is based on the $CO_2$ proxy method, which use the ratio between simulated ($XCO_2^{model}$) and retrieved ($XCO_2^{raw}$) $CO_2$ columns to correct for retrieved methane columns ($XCH_4^{raw}$) that are sensitive to aerosol and surface interference:

$$XCH_4 = \frac{XCH_4^{raw}}{XCO_2^{raw}} \times XCO_2^{model} \tag{1}$$

This limits variable biases because both $XCH_4^{raw}$ and $XCO_2^{raw}$ are similarly affected by scattering artefacts, but the method is subject to any biases in specified $CO_2$ columns (Parker et al., 2015).

The University of Leicester Proxy $XCH_4$ v9.0 retrieval takes the median $CO_2$ columns from three atmospheric chemistry transport models as $XCO_2^{model}$, and the range of the three models characterizes the $XCO_2^{model}$ uncertainty (Figure S1). The disagreement among these three models is ~1 ppm over remote regions, ~2 ppm over East China, and 2–4 ppm in India, Bangladesh, and southwestern China, which translates roughly to uncertainties of 0.3, 0.4, and 0.5–1.0%, respectively, in retrieved $XCH_4$.

For TROPOMI, we use the SRON RemoTeC-S5P $XCH_4$ scientific product, from Lorente et al. (2021). The improved algorithm by Lorente et al. (2021) was later implemented in the official operational product (v2.02.00) in July 2021 (Lorente et al., 2022). They derived an empirical correction formula to improve surface reflectance dependent biases identified in TROPOMI full-physics retrievals. The correction in general improves data quality over scenes with low (e.g., snow cover) and high surface albedo (e.g., deserts) which are challenging for a full-physics algorithm. Large corrections are made in East China, Xinjiang China, Southeast Asia, and Siberia (Figure S2). Bias-corrected TROPOMI retrievals flagged with "qa_value = 1" are used for inversion. This version of the TROPOMI product does not provide ocean glint-mode retrievals.

Figure 1 shows the spatial distributions of $XCH_4$ measured by GOSAT and TROPOMI, annually averaged on the $0.625° \times 0.5°$ grid. Both datasets show high $XCH_4$ in eastern China and northern India and low $XCH_4$ over the Mongolian and Tibetan plateaus, although TROPOMI provides much better spatial coverage than GOSAT over most regions. There are in total 45,018

observations for GOSAT and 8,860,722 for TROPOMI. We take averages when multiple measurements fall within a 0.625° × 0.5° grid cell on any individual day (this procedure affects primarily dense TROPOMI data), and the resulting gridded daily

observations are used in the inversion. The spatial distribution of gridded daily observation numbers is shown in Figure S3.

We refer to the XCH$_4$ retrieval products used in this study as GOSAT or TROPOMI observations and corresponding inversions as GOSAT or TROPOMI inversions for simplicity. There are other operational and scientific retrieval products available from both GOSAT and TROPOMI measurements (e.g., the operational GOSAT XCH$_4$ retrieval (Yoshida et al., 2013), the scientific

TROPOMI/WFMD XCH$_4$ retrieval (Schneising et al., 2019)). Our analyses and conclusions are specific to the two retrieval products used here, though we expect that some of them can also apply to other retrievals.

## XCH$_4$ observations from two different satellite instruments

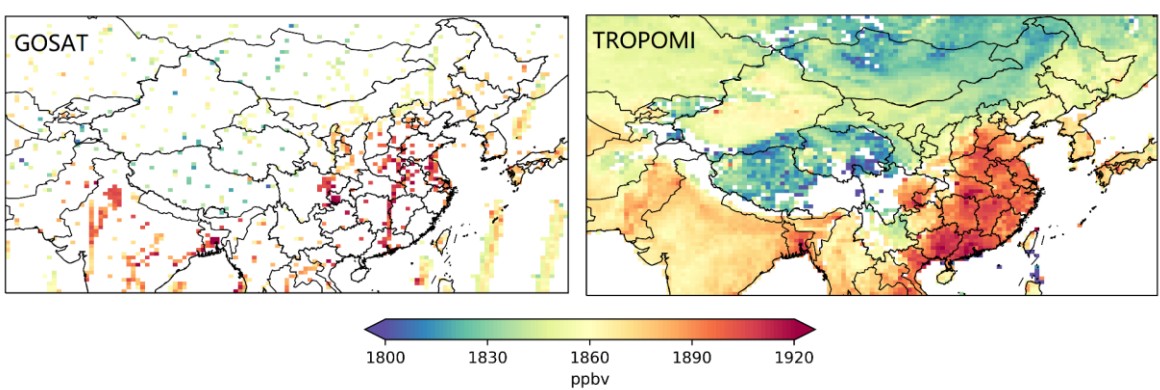

**Figure 1: 2019 annual average methane column mole fractions over the East Asia domain for GOSAT (UoL proxy v9.0 retrieval) and TROPOMI (Lorente et al. (2021) full-physics retrieval), presented on the 0.5° × 0.625° GEOS-Chem grid.**

**2.2 Independent evaluation data**

We use a suite of independent high-quality methane observations to evaluate the posterior emissions inferred from satellite observations, including surface *in situ* observations, ground-based remote sensing observations, and tropospheric *in situ* measurements from commercial airlines. Table S1 provides a descriptive list of these surface sites and Figure 2 shows the locations of surface sites and a representative flight path. These suborbital observations are of good accuracy and precision

compared to satellite observations.

Surface *in situ* observations are available through World Data Centre for Greenhouse Gases (WDCGG) or the CH$_4$ GLOBALVIEWplus v4.0 ObsPack (Schuldt et al., 2021). The five sites are Anmyeon-do, South Korea (AMY), Pha Din, Vietnam (PDI), Lulin, Taiwan China (LLN), Ulaan Uul, Mongolia (UUM), Waliguan, China (WLG) (Dlugokencky et al.,

1994; Dlugokencky et al., 2021; Lee et al., 2019; Nguyen Nhat Anh and Steinbacher, 2021). Observations are done with either continuous (hourly) online instruments or weekly collected flask (Table S1). Daytime measurements are used for evaluating

simulations. Most of these sites are continental or subcontinental background sites (PDI, LLN, UUM, and WLG), and their observations are insensitive to local methane emissions. An exception is AMY which is affected by local Korean emissions as well as upwind East China emissions.


Total methane column observations by ground-based Fourier Transform Spectrometers are available at two TCCON sites located in East China, Hefei, China (HF) and Xianghe, China (XH) (Liu et al., 2023; Yang et al., 2020), and their observations are sensitive to methane emissions from East China. We note that a previous evaluation of GOSAT and TROPOMI against TCCON did not include data from these two sites, as their data were not available then (Qu et al., 2021). We use only
measurements with solar zenith angles < 60° to ensure high data quality.

All the above surface sites are far from northern India, which is a major methane emitting region in the study domain. The only relevant dataset available to us in this area comes from the Civil Aircraft for the Regular Investigation of the atmosphere Based on an Instrument Container (CARIBIC) project (available via the CH$_4$ GLOBALVIEWplus v4.0 ObsPack (Schuldt et
al., 2021)), which includes regular flights in the troposphere over northern India. However, these data are collected in earlier years between 2012 and 2014 before the time of TROPOMI. In the absence of better observation data, we compare these 2012–2014 aircraft observations to a simulation driven by a similarly configured GOSAT inversion for an earlier period (2010–2017) (Zhang et al., 2022). By doing so, we assume that any systematic bias derived from this comparison should still be representative of the 2019 GOSAT inversion.

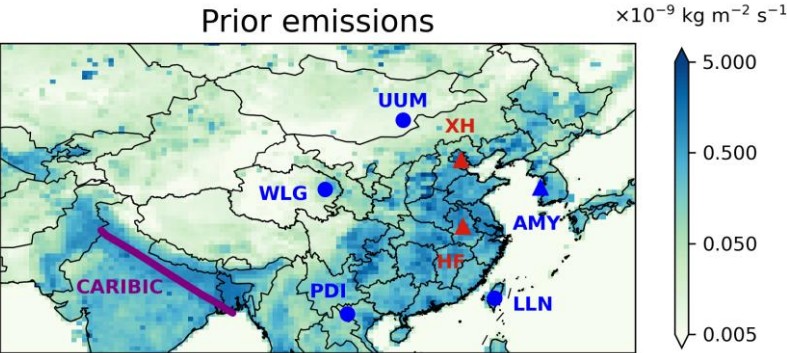


**Figure 2: Spatial distribution of prior emissions. Locations of independent data for evaluation (seven surface sites and aircraft route) are shown. Circles represent background sites and triangles source-region sites. Total column measurements are coded in red and *in situ* measurements in blue. Purple solid line shows a CARIBIC aircraft route that measured tropospheric methane over India on November 22, 2012.**

## 3 Inverse analysis

### 3.1 Forward model and prior emissions

We use GEOS-Chem v12.9.3 as the forward model for the inversion. The simulation is conducted for 2019 over East Asia (15°N–55°N, 60°E–140°E) on a 0.5° × 0.625° horizontal grid with 47 vertical layers and is driven by MERRA-2 meteorological fields from the NASA Global Modeling and Assimilation Office (GMAO) (Gelaro et al., 2017). The initial concentration fields on January, 1, 2019 and 3-hourly boundary conditions for the nested domain are taken from a global inversion of TROPOMI data for 2019 (Qu et al., 2021). We find that the boundary conditions from this global inversion still have biases over East Asia (discussed further in Section 4.3.3), which may partly be due to the fact that Qu et al. (2021) used an earlier version of TROPOMI retrievals. In our inversion, we optimize for systematic biases at four lateral boundaries together with methane emissions.

Prior emissions (Figure 2) used in GEOS-Chem simulations are compiled from bottom-up sectoral inventories (Table S2). In brief, we use EDGAR v4.3.2 (Janssen-Maenhout et al., 2019) for anthropogenic methane emissions, with those from fossil fuel exploitation replaced by Scarpelli et al. (2020) (oil and gas; coal outside of China) and Sheng et al. (2019) (coal in China). A comparison with a more recent inventory EDGAR v6 shows no large revisions of anthropogenic methane emissions over the study region that we expect to have a great impact on the inversion results (Figure S4). For natural emissions, we use the ensemble average of the WetCHARTs version 1.0 inventory for wetlands (Bloom et al., 2017), the Quick Fire Emissions Dataset (QFED) v2.4r8 for biomass burning, Fung et al. (1991) for termite emissions, and Maasakkers et al. (2019) for geological sources.

While methane sinks are not optimized in our regional inversion, they are explicitly simulated in GEOS-Chem simulations. We use monthly OH fields from a full-chemistry GEOS-Chem simulation (Wecht et al., 2014) and soil absorption from Murguia-Flores et al. (2018).

### 3.2 Inversion procedure

We perform analytical Bayesian inversions to optimize a state vector $x$ containing annual methane emissions from 600 clusters and average methane column biases at four model boundaries. We optimize emissions on 600 spatial clusters instead of the native 0.5° × 0.625° grid (Figure S5), which are generated based on a Gaussian Mixed Model (GMM) algorithm proposed by Turner and Jacob (2015). This strategy significantly reduces the computation of an analytical inversion while accounting for major patterns in the distribution of methane emissions. We also optimize for biases in boundary conditions on four sides of our domain (east, south, west, north). Examination of our prior simulation finds domain-wide biases against either GOSAT or TROPOMI observations that can only be attributed to biased boundary conditions. The optimization is done annually for our main result. In addition, we also perform a seasonal optimization in a sensitivity inversion.

Assuming a Gaussian distribution of error, the optimal estimate of $x$ is obtained by minimizing the cost function (Brasseur and Jacob, 2017; Rodgers, 2000):

$$J(x) = (x - x_A)^T S_A^{-1}(x - x_A) + \left(y - F(x)\right)^T S_O^{-1}(y - F(x)) \tag{2}$$

where $x_A$ is prior estimates for $x$ and $y$ is the observation vector containing either TROPOMI or GOSAT observations, and $F$ is a function of $x$ representing the forward model. $S_A$ and $S_O$ are respectively prior and observation error covariance matrices, and their specification is described and discussed in Section 3.3.

The forward model (GEOS-Chem) can be described by a linear equation:

$$F(x) = Kx, \tag{3}$$

where $K = \nabla_x F$ is the Jacobian matrix, which describes the sensitivity of observations to the state vector. The cost function is minimized at $\nabla_x J(x) = 0$, which yields the optimal estimate ($\hat{x}$)

$$\hat{x} = x_A + (K^T S_O^{-1} K + S_A^{-1})^{-1} K^T S_O^{-1}(y - Kx_A), \tag{4}$$

with the posterior error covariance matrix $\hat{S}$

$$\hat{S} = (K^T S_O^{-1} K + S_A^{-1})^{-1} \tag{5}$$

and the averaging kernel matrix $A$ that describes the sensitivity of the optimal solution to the true value:

$$A = \frac{\partial \hat{x}}{\partial x} = I_n - \hat{S} S_A^{-1}. \tag{6}$$

The trace of $A$ is referred to as the degree of freedom for signals (DOFS), which represents the number of independent pieces of information constrained by an observing system.

We apply a transformation vector $w$ to aggregate the posterior estimate regionally ($\hat{x}_r = w^T \hat{x}$). The corresponding posterior error covariance for the region ($\hat{\sigma}_r^2$) is then computed as

$$\hat{\sigma}_r^2 = w^T \hat{S} w. \tag{7}$$

### 3.3 Error specification

The observation error covariance matrix $S_O$ represent total random errors from both the methane retrieval ($y$) and the forward model ($F(x)$). It can be decomposed as $S_O = \Sigma C \Sigma$, where $\Sigma$ is the diagonal standard deviation matrix and $C$ is the error correlation matrix. In general, inverting $S_O$ (which has a dimension of 10,000–10,000,000) in Eq. (4) and (5) is computationally difficult if $C$ is non-diagonal. The computational challenge can be eased by omitting error correlations ($S_O = \Sigma^2$), but this assumption of error independence unrealistically increases the power of individual observations leading to overfitting (highly unlikely departure of the posterior solution from the prior estimate) (Zhang et al., 2018). To remedy this issue, previous studies

introduce a scalar factor $\gamma$ ($\mathbf{S_O} = \frac{\Sigma^2}{\gamma}$), which serves to enlarge the observation error ($\gamma$ is usually $< 1$) and thus de-weight individual observations (e.g., Maasakkers et al., 2019; Qu et al., 2021; Zhang et al., 2018). The $\gamma$ value, which plays the same role as the regularization parameter in Tikhonov methods, can be determined through the graph-based L-curve method (Hansen, 1998; Lu et al., 2021); however, results are sometimes ambiguous and often difficult to interpret physically.

Here, we propose an alternative method. We first determine the diagonal matrix $\Sigma$ following the residual error method (Heald et al., 2004), which yields observation error standard deviations averaged 16 ppbv for TROPOMI and 18 ppbv for GOSAT, respectively. Then, we specify a full error correlation matrix $\mathbf{C}$. We parametrize the entry $C_{ij}$ as a function of the distance ($\Delta d_{ij}$) and the time ($\Delta t_{ij}$) between $i^{th}$ and $j^{th}$ observations:

$$C_{ij} = \exp\left(-\frac{\Delta d_{ij}}{\rho_d}\right)\exp\left(-\frac{\Delta t_{ij}}{\rho_t}\right), \tag{8}$$

where $\rho_d$ and $\rho_t$ are correlation scales in space and time, respectively. The values of $\rho_d$ and $\rho_t$ can be determined empirically by analyzing spatial and temporal correlations in prior residual errors (Figure S6). In our case, we find $\rho_t = 7$ days and $\rho_d = 400$ km. Finally, we find $\tilde{\mathbf{C}}^{-1}$, a computationally tractable (diagonal) approximation to $\mathbf{C}^{-1}$, and replace $\mathbf{S_O}^{-1}$ in Eq. (4) and (5) with $\Sigma^{-1}\tilde{\mathbf{C}}^{-1}\Sigma^{-1}$. See Appendix A for the derivation of $\tilde{\mathbf{C}}^{-1}$. Compared to the traditional $\gamma$ factor, this method provides better interpretability by explicitly specifying error correlations. Moreover, $\tilde{\mathbf{C}}^{-1}$ can be unequivocally determined once $\mathbf{C}$ is specified. For comparison, we also include a sensitivity inversion in which $\mathbf{S_O}$ is specified as $\frac{\Sigma^2}{\gamma}$ with $\gamma = 0.6$ for GOSAT observations and $\gamma = 0.09$ for TROPOMI observations following the procedure by Lu et al. (2021) (Figure S7).

For the prior error covariance matrix $\mathbf{S_A}$, we take it as a diagonal matrix and assume a 50% standard deviation for prior emissions and a 1% standard deviation for boundary conditions. We also test two alternative configurations for $\mathbf{S_A}$ in sensitivity inversions: (1) the relative error standard deviation for prior emissions is enlarged to 100%; (2) the error standard deviation for prior emissions is specified as 50% or $1\times10^{-10}$ kg m$^{-2}$ s$^{-1}$ whichever is larger. The latter $\mathbf{S_A}$ specification gives the inversion more freedom to adjust at locations where prior methane emissions are small or none.

## 4 Results and discussion

### 4.1 Comparison of methane emissions from TROPOMI and GOSAT inversions

Figure 3 shows the correction patterns of methane emissions (posterior−prior emissions) inferred respectively from TROPOMI and GOSAT inversions. Both inversions find that the prior inventory underestimates methane emissions from Northeast China (NEC) and Bangladesh (BAN) and overestimates emissions from Central South China (CSC). The two inversions also find similar correction patterns in Central North China (CNC) with upward adjustments over central Shanxi and downward

adjustments over neighboring Henan province. These agreements reflect some consistencies between TROPOMI and GOSAT inversions at the regional level.

TROPOMI and GOSAT inversions show large differences over important source regions, including East China (EC) and northern India (IND) (Figure 3). While the GOSAT inversion suggests that methane emissions over IND should be increased and those from EC decreased relative to prior estimates, the TROPOMI inversion finds the opposite. As a result, regional total methane emissions inferred by the two inversions differ by 7.7 Tg a$^{-1}$ or 27% over IND (TROPOMI: 24.6 ± 0.6 Tg a$^{-1}$, GOSAT: 32.3 ± 0.8 Tg a$^{-1}$) (errors reported for regional estimates are 1$\sigma$ standard deviations derived from posterior error covariance

matrices) and by 6.4 Tg a$^{-1}$ or 29% over EC (TROPOMI: 28.0 ± 0.8 Tg a$^{-1}$, GOSAT: 21.6 ± 1.0 Tg a$^{-1}$) (Figure 3c). In addition, the two inversions also disagree over the northwestern part of the domain (NWD including parts of Kazakhstan and northern Xinjiang, China and SXJC including mainly southern Xinjiang, China), where TROPOMI indicates large upward adjustments while GOSAT finds agreement with the prior inventory.

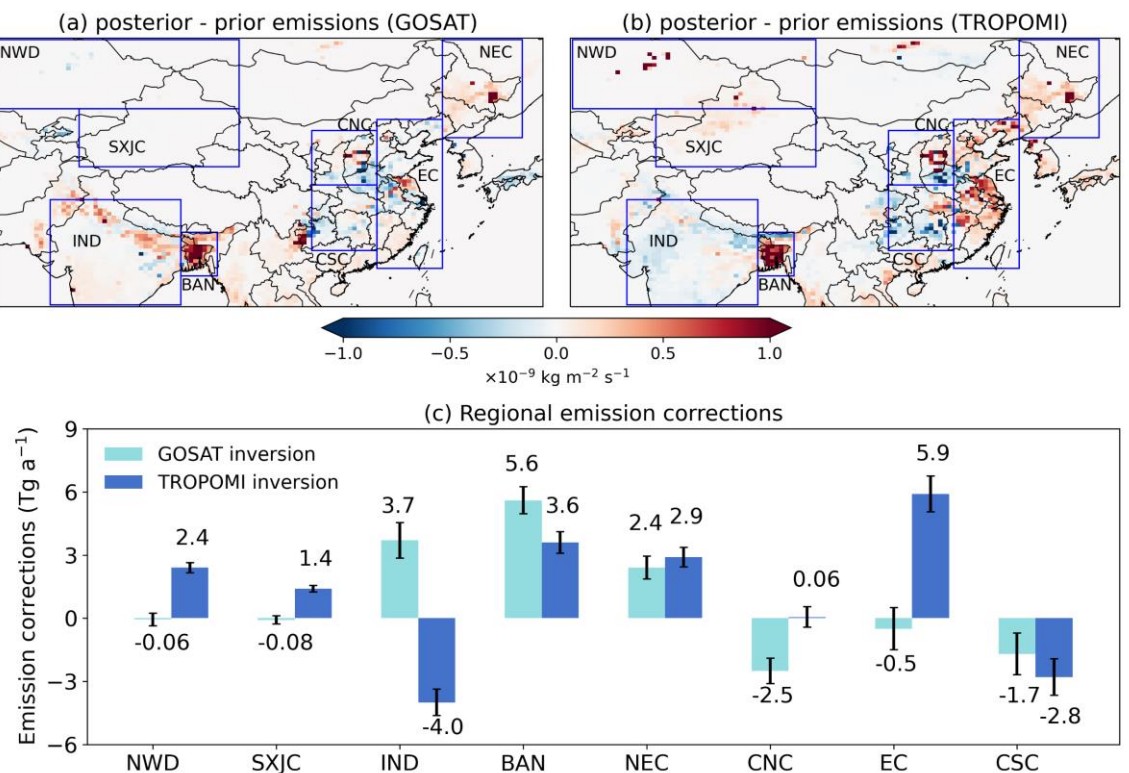

Figure 3: Spatial distributions of methane emission corrections (posterior−prior) inferred by (a) GOSAT and (b) TROPOMI inversions. (c) shows emissions aggregated by region as defined in blue rectangles in (a) and (b). Error bars represent the standard deviation of regional estimates derived from posterior error covariance matrices (Eq. 7). These errors do not include systematic uncertainties due to inversion setups and are thus optimistic, but they are relevant for comparing results from two identically configured inversions.


Table S2 summarizes methane emission estimates from TROPOMI and GOSAT inversions over the entire East Asia domain and over China. The two inversions find consistent posterior methane emissions from East Asia (TROPOMI: $142.7 \pm 1.3$ Tg $a^{-1}$; GOSAT: $142.6 \pm 1.5$ Tg $a^{-1}$), with differences in China (TROPOMI: $73.7 \pm 0.9$ Tg $a^{-1}$; GOSAT: $66.4 \pm 1.1$ Tg $a^{-1}$) largely canceled out by differences in northern India. For China, we attribute 67.9 Tg $a^{-1}$ for the TROPOMI inversion and 61.6 Tg $a^{-1}$

for the GOSAT inversion to anthropogenic emissions, based on prior sectoral fractions in each spatial cluster. These values are at the high end of previous inversion-based estimates of 43–62 Tg $a^{-1}$ (Deng et al., 2022; Lu et al., 2021; Miller et al., 2019; Qu et al., 2021; Saunois et al., 2020; Sheng et al., 2021; Stavert et al., 2022; Wang et al., 2021; Zhang et al., 2021; Zhang et al., 2022) and are higher than China's latest submission to the UNFCCC (55 Tg $a^{-1}$) for 2014 (UNFCCC, 2020). These previous inversions mainly used GOSAT observations but differ greatly in their inversion setups (e.g., time, domain coverage, spatial

resolution, transport model), thus resulting in a considerable range of estimates. In contrast, the differences in inversions presented in this work are fully due to satellite observations. Our TROPOMI inversion results are consistent with a recent TROPOMI inversion study by Chen et al. (2022) who reported estimates of China's total, anthropogenic, and natural methane emissions of 70.0 (61.6–79.9), 65.0 (57.7–68.4), and 5.0 (3.9–11.6) Tg $a^{-1}$.

In addition to the main inversion, we also perform a series of sensitivity inversions. The objective is to test whether the comparison between the GOSAT and TROPOMI inversions (e.g., Figure 3) is affected by the configurations such as the assimilation window and error specifications. There are 4 sensitivity tests including (1) optimizing emissions seasonally instead of annually; (2) increasing prior error standard deviations from 50% to 100%; (3) assigning a minimum prior error standard deviation equivalent to $1\times10^{-10}$ kg m$^{-2}$ s$^{-1}$; and (4) applying a traditional regularization factor following Lu et al. (2021) ($\gamma =$

0.6 for GOSAT and $\gamma = 0.09$ for TROPOMI) to account for error correlations in $\mathbf{S_0}$ instead of the method proposed in Section 3.3.

Figure S8 and S9 show that the major findings from our comparison of the GOSAT and TROPOMI inversions shown in Figure 3 (agreement in NEC, BAN, and CSC; disagreement in EC, IND, NWD, SXJC) are robust against these perturbed inversion

configurations. Consistent with the main inversion, the sensitivity tests find good agreement between the GOSAT and TROPOMI inversions in posterior methane emissions from NEC (upward adjustment), BAN (upward adjustment), and CSC (downward adjustment), but find that discrepancies range 4.1–8.2 Tg $a^{-1}$ for EC and 5.1–8.8 Tg $a^{-1}$ for IND (Figure S9). These results indicate that the effects of inversion configurations are only moderate on systematic differences between the GOSAT and TROPOMI inversions.


**4.2 Evaluation of inversion results with independent observations**

Both TROPOMI and GOSAT posterior simulations can reduce errors against their respective "training" data relative to the prior simulation (Figure 4), which is expected for successful inversions. However, concentration fields from the two simulations show varied degrees of agreement across the domain (Figure 5a). In this section, we use independent high-quality observations to evaluate whether GOSAT and TROPOMI inversion results are consistent, and in the case that they are not, which one is in better agreement with independent data.

Table 1 summarizes performance metrics against these independent observations. Figure S10 plots the timeseries of these observations in comparison with prior and posterior simulations. GOSAT and TROPOMI inversions perform similarly at background sites such as PDI, UUM, WLG, and LLN. Both posterior simulations considerably reduce biases against *in situ* observations at WLG and PDI and achieve reasonable agreement at PDI, UUM, and WLG (absolute biases < 8 ppbv and $R^2$ between 0.40–0.73). Among these sites, WLG show a relatively low posterior $R^2$ (GOSAT: 0.40; TROPOMI: 0.41) due to inability to capture sub-seasonal variability (Figure S10). Seasonal optimization done in one of the sensitivity inversions only improves $R^2$ at WLG marginally (Figure S10). An exception is LLN (a high-mountain background site in the southeast of the domain) where biases grow larger in both posterior simulations (10.8 ppbv for GOSAT and 16.7 ppbv for TROPOMI). This is mainly caused by large seasonal biases in the eastern boundary (Figure 5c) (see Section 4.3.3 for more discussion). The bias is the largest during the monsoon season (May to August) (Figure S10).

On the other hand, methane concentrations from the TROPOMI and GOSAT posterior simulations differ by ~10–20 ppbv at sites in methane source regions (i.e., XH and HF within EC and AMY in Korea downwind EC) (Figure 5a). Their differences in concentrations are due mainly to higher methane emissions inferred by the TROPOMI inversion than GOSAT over EC (by 6.4 Tg a$^{-1}$) and Korea (Figure 3). Our evaluation against *in situ* measurements at AMY and total column measurements at XH and HF shows consistently high biases of ~15–27 ppbv by the TROPOMI posterior simulation and a comparatively better agreement (bias ~8 ppbv) with the GOSAT posterior simulation (Table 1). Smaller mean biases are achieved by the prior simulation at XH and HF (Table 1), but this is largely because of the low background concentration caused by biases in prior boundary conditions (as indicated by the large negative prior bias at the upwind background site WLG; Table 1). The ability to capture temporal variations can be further improved by seasonal optimization of emissions, especially for HF where the influence of the seasonal cycles in rice emissions is strong (Figure S10). Overall, our results at AMY, XH, and HF supports the lower methane emissions from EC inferred by the GOSAT inversion over the TROPOMI inferences and indicates that TROPOMI XCH4 retrievals may have regional high biases over EC (more discussion in 4.3.1).

Methane concentrations from the TROPOMI and GOSAT posterior simulations differ by 4.9 ppbv on average along the CARIBIC flight tracks over the Indo-Gangetic Plain (Figure 5a). This difference is mainly due to different IND methane

emissions between the two inversions (Figure 5b) with minor contributions from boundary condition bias inferences (Figure 5c). In the absence of concurrent independent observations over IND, we use CARIBIC aircraft observations that are only available from 2012 to 2014 to evaluate the inversions. Since these observations predate TROPOMI, we can only indirectly evaluate by using a simulation driven by methane emissions from a GOSAT inversion for earlier years as an inter-comparison platform. We take inversion results from a previous study (Zhang et al., 2022), which performed an East Asia inversion also using GOSAT proxy XCH4 retrievals. Their inversion is almost identically configured as this study except that it was for 2010–2017. Consistent with our GOSAT results, the GOSAT inversion from Zhang et al. (2022) also found that IND methane emissions should be adjusted upward.

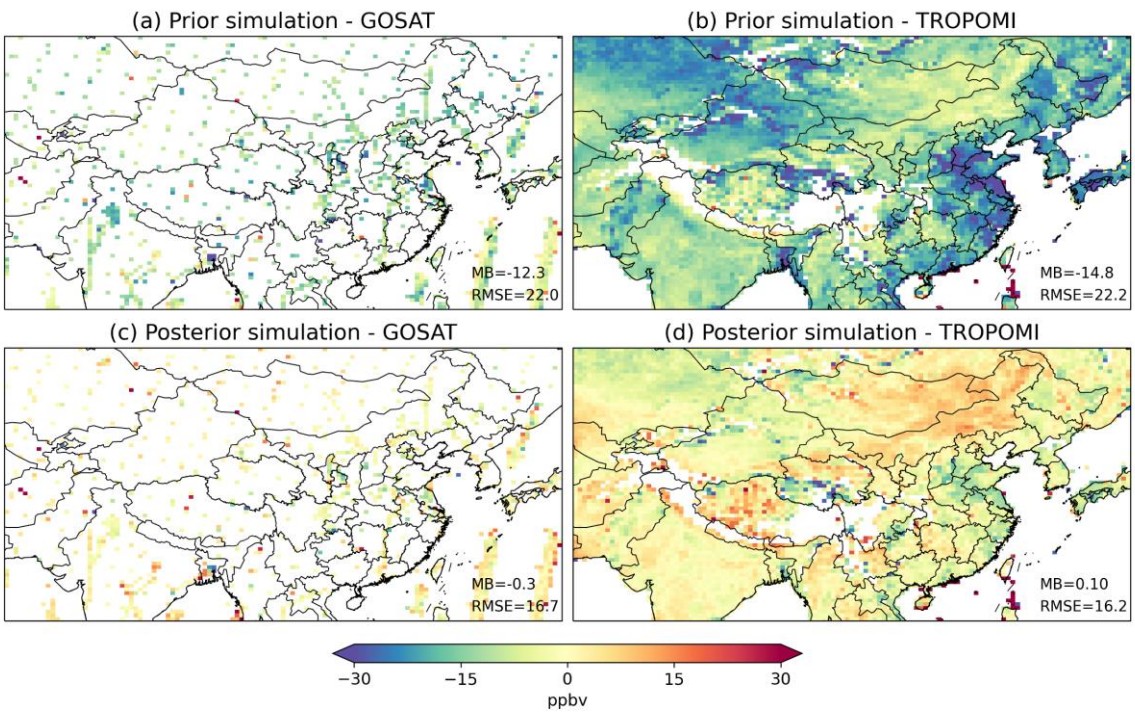

Figure 4: Differences in XCH4 between simulations and satellite observations from GOSAT (a and c) and TROPOMI (b and d). (a) and (b) show results for the prior simulation, (c) for the posterior simulation driven by the GOSAT inversion, and (d) for the posterior simulation driven by the TROPOM inversion. Root-mean-square errors (RMSE, in ppbv) and mean biases (MB, in ppbv, simulation − observation) are inset.

**Table 1: Evaluation of simulated methane concentrations against independent observations[a].**

| Site | Mean Bias ± Standard Error (ppbv) | | | $R^{2,b}$ | | |
|---|---|---|---|---|---|---|
| | Prior | GOSAT | TROPOMI | Prior | GOSAT | TROPOMI |
| **AMY** | −5.9 ± 2.5 | 7.9 ± 2.4 | 27.2 ± 2.7 | 0.46 | 0.50 | 0.46 |
| **PDI** | −20 ± 2.3 | −5.5 ± 2.2 | −1.4 ± 2.2 | 0.67 | 0.70 | 0.70 |
| **LLN** | 0.5 ± 4.1 | 10.8 ± 4.2 | 16.7 ± 4.3 | 0.39 | 0.40 | 0.37 |
| **UUM** | −9.0 ± 2.1 | 6.3 ± 2.0 | 7.8 ± 2.2 | 0.71 | 0.73 | 0.72 |
| **WLG** | −16.6 ± 2.7 | −4.1 ± 2.7 | −2.5 ± 2.5 | 0.40 | 0.40 | 0.41 |
| **XH[c]** | −3.4 ± 1.0 | 8.4 ± 0.9 | 15.3 ± 1.0 | 0.72 | 0.75 | 0.74 |
| **HF[c,d]** | 1.0 ± 3.0 | 9.0 ± 3.1 | 20.6 ± 3.2 | 0.53 | 0.53 | 0.57 |
| **CARIBIC[e]** | – | 14.9 ± 0.8 | 10.0 ± 0.8 | – | – | – |

[a] Five sites report surface *in situ* measurements with PDI, LLN, UUM, and WLG being continental-scale background sites and AMY a regional site. Two sites (XH and HF) located in East China report ground-based total column measurements. The aircraft measurements (CARIBIC) are taken over northern India.

[b] Main inversions are unable to improve the performance for temporal variability, as the optimization of methane emissions is done only annually. Seasonal inversions improve the performance at site AMY, WLG, XH, LLN (only GOSAT), and HF but in most cases only slightly. Other factors that affect the $R^2$ metric include model transport errors and observation representativeness.

[c] Small prior biases at XH and HF should not be interpreted as evidence for unbiased prior emissions from EC, because the prior simulation has substantial low biases in background concentrations as shown by data at WLG (upwind of EC).

[d] Large biases between simulations and observations occur in five days (Jul. 22rd, Sept. 30th, Nov. 3rd, Nov. 23rd and Dec. 3rd) at site HF (Figure S10). Relatively low $R^2$ in this line are largely affected by these data. Excluding this subset of observations results in correlation coefficients of ~0.8 for all simulations and mean biases of −2.9 ± 1.4, 5.4 ± 1.3, and 15.9 ± 1.7 ppbv for prior, GOSAT, and TROPOMI simulations, respectively.

[e] Indirect evaluation is performed for CARIBIC data. The value in the 'GOSAT' column represents the mean bias between the posterior simulation of the 2010–2017 GOSAT inversion and 2012–2014 CARIBIC aircraft observations. We assume that GOSAT inversions are consistent between years so that the 2012–2014 bias is representative for the 2019 condition. The value in the 'TROPOMI' column is computed by subtracting the mean difference along aircraft paths between 2019 GOSAT and TROPOMI posterior simulations (~4.9 ppbv) (Figure 5a) from the 2012–2014 GOSAT bias. $R^2$ is not reported for this indirect comparison.

Comparison with these aircraft observations indicates that the 2012–2014 simulation driven by GOSAT-optimized emissions from Zhang et al. (2022) overestimates the aircraft observations by ~14.9 ppbv (Table 1). On the other hand, the 2019 posterior simulation from the GOSAT inversion is about 4.9 ppbv higher than that from the TROPOMI inversion along flight tracks (Figure 5a). Assuming that our 2019 GOSAT inversion is consistent with the 2010–2017 GOSAT inversion by Zhang et al. (2022) (mean bias 14.9 ppbv), it thus suggests that the TROPOMI inversion likely agrees better with the CARIBIC

observations (mean bias 10.0 ppbv) than the GOSAT inversion. Unlike the EC case, we find over IND relatively small
systematic differences in TROPOMI and GOSAT XCH$_4$ retrievals (Figure 6). Our analysis suggests that good data coverage
of TROPOMI over IND is likely responsible for its better performance in constraining methane emissions (see section 4.3.2
for more discussion).

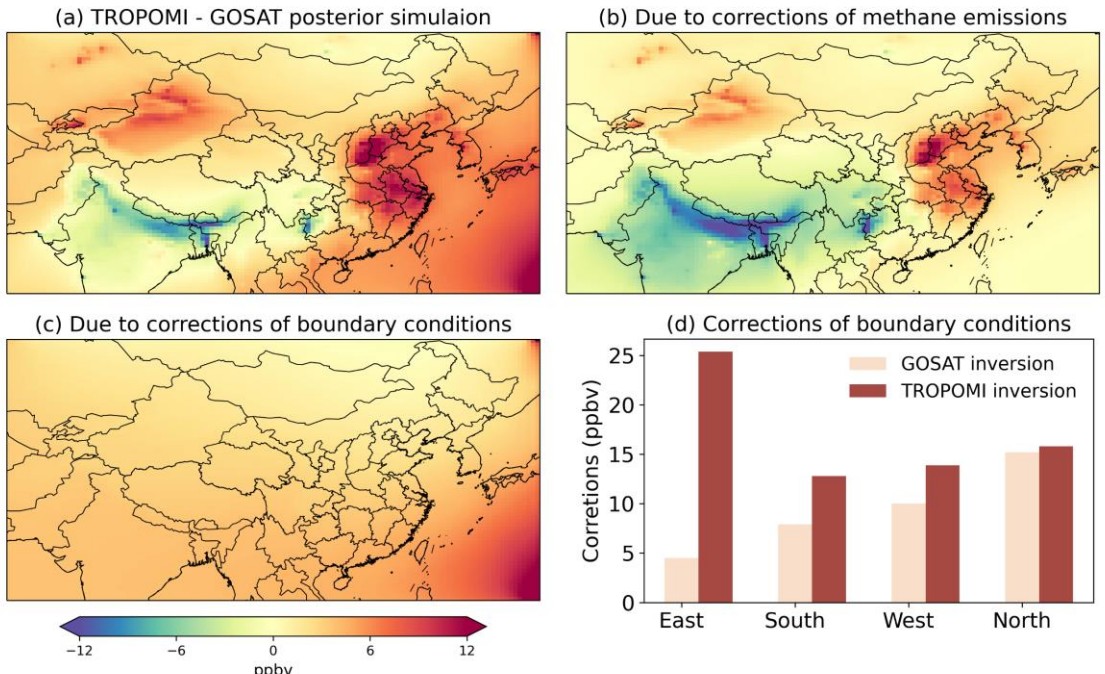

**Figure 5: Differences in tropospheric methane concentrations (TROPOMI − GOSAT) between GOSAT and TROPOMI posterior simulations. (a) shows the total differences while (b) and (c) decompose the differences to methane emissions and boundary condition bias corrections. The corrections of boundary conditions (in ppbv) by the two inversions are shown.**

## 4.3 Attribution of TROPOMI and GOSAT inversion differences

### 4.3.1 Regional differences in XCH$_4$ retrievals

To understand the cause of differences in the inferred methane emissions, we first compare coincident TROPOMI and GOSAT
XCH$_4$ retrievals. The comparison is done following Zhang et al. (2010) where a CTM simulation is used as an intercomparison
platform to account for differences in prior profiles and vertical sensitivity between TROPOMI and GOSAT retrievals.
TROPOMI XCH$_4$ are on average higher than GOSAT XCH$_4$ over EC by ~6 ppbv, SXJC by ~10 ppbv, and NWD by ~10 ppbv
(Figure 6b), which lead to higher methane emissions inferred by the TROPOMI inversion over these regions (Figure 3). These
differences persist throughout the year in EC and SXJC but appear to be highly seasonal in NWD. The largest TROPOMI-
GOSAT differences in NWD (~30–40 ppbv) occur between December and March. In other regions of interest, the annual
averaged TROPOMI-GOSAT XCH$_4$ differences are in general less than 5 ppbv including IND where the two inversions find
large discrepancies in posterior methane emissions.

Independent ground-based observations are more consistent with the GOSAT inversion and thus do not support high emissions

from EC inferred by the TROPOMI inversion, which indicates that TROPOMI retrievals have systematic regional high biases over EC. In addition, even with enhanced methane emissions in EC, SXJC, and NWD from the TROPOMI inversion, the posterior simulation cannot fully capture these high $XCH_4$ concentrations (Figure 4d). This is also a hint of retrieval biases, as it indicates that the inversion finds it difficult to reconcile these high $XCH_4$ patterns with known methane sources and wind information, given our specification of error parameters ($\mathbf{S_A}$ and $\mathbf{S_O}$).

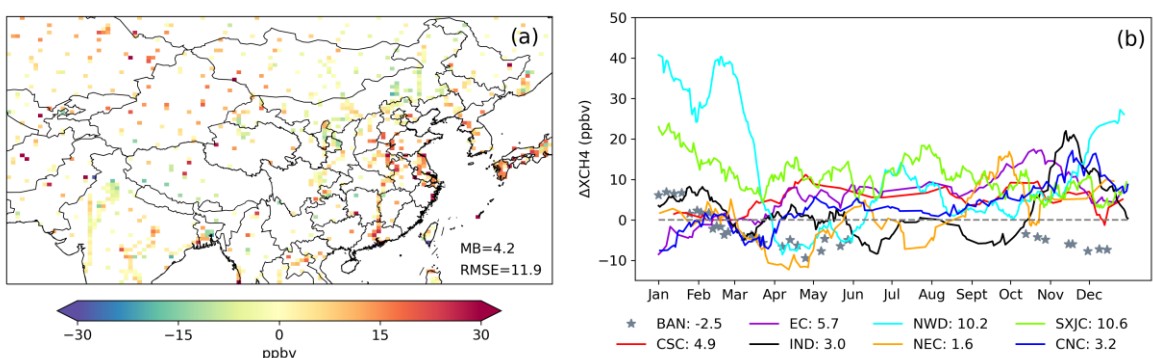


**Figure 6: Differences in $XCH_4$ between GOSAT and TROPOMI (defined as TROPOMI − GOSAT) shown on the $0.5° \times 0.625°$ grid (a) and by region (b). (a) shows annual averages for each grid cell and (b) shows time series of regional averages. Regions are defined in blue rectangles of Figure 3a.**

In addition to EC, large $XCH_4$ differences between GOSAT and TROPOMI are also found in the northwestern part of the

domain (SXJC and NWD). Although we do not have independent observations over these regions, we speculate that TROPOMI retrievals have positive biases. SXJC is featured with high surface albedo (desert), while in NWD large TROPOMI and GOSAT differences occur during Dec and Mar when surface albedo is low (snow and/or ice cover) (Figure S11). High and low surface albedo scenes are known to be challenging for the full-physics retrieval. We suggest to apply the "blended albedo" filter to TROPOMI observations over these regions before inversion (Chen et al., 2022; Wunch et al., 2011).


In our study, we use the TROPOMI science product from Lorente et al. (2021), who applied a posterior correction for surface albedo dependent biases identified in originally retrieved TROPOMI data. We find that this bias correction scheme does overall improve the agreement between TROPOMI and GOSAT in both their methane column concentrations (Figure S12) and posterior methane emissions (Figure S13). However, the agreement is not improved in EC, SXJC, and NWD.


Previous studies have reported decreased accuracy of GOSAT $CO_2$ proxy retrievals in India owing to errors in the specified $CO_2$ field (Parker et al., 2015; Schepers et al., 2012), which is consistent with a large uncertainty in modeled $XCO_2$ applied to GOSAT $CH_4$ retrievals in India (Figure S1). The range of modeled $XCO_2$ used in the GOSAT product is equivalent to an $XCH_4$

uncertainty of 0.7% (~ 13 ppbv) in India and Bangladesh.   Our result shows that TROPOMI XCH$_4$ is lower than GOSAT

XCH$_4$ in the western Indo-Gangetic Plain (around Delhi) and higher in a few locations outside the Indo-Gangetic Plain (Figure

6a), but the regional difference between the two retrievals is overall small (< 5 ppbv) in IND compared to those in EC, SXJC,

and NWD (Figure 6b). Exceptions are November and December when the differences are up to 20 ppbv in IND.

### 4.3.2 Spatial coverage of observations

Although methane emissions from IND inferred by the GOSAT inversion are considerably larger than those inferred by the

TROPOMI inversion, we find relatively small differences in coincident XCH$_4$ retrievals there (Figure 6), indicating that

retrieval biases are unlikely the only cause of discrepancies. Moreover, the two satellite products differ greatly in their data

density over the Indo-Gangetic Plain (blue ellipse in Figure 7) where the discrepancy in inferred methane emissions is the

largest. GOSAT has almost no observations over the region, while TROPOMI samples the region fairly well (Figure S3). We

have shown above that indirect comparison with CARIBIC tropospheric aircraft measurements favors lower emissions from

IND estimated by the TROPOMI inversion (Table 1). In this section, we explore whether differences in data coverage between

TROPOMI and GOSAT may contribute to the discrepancies in inferred emissions.

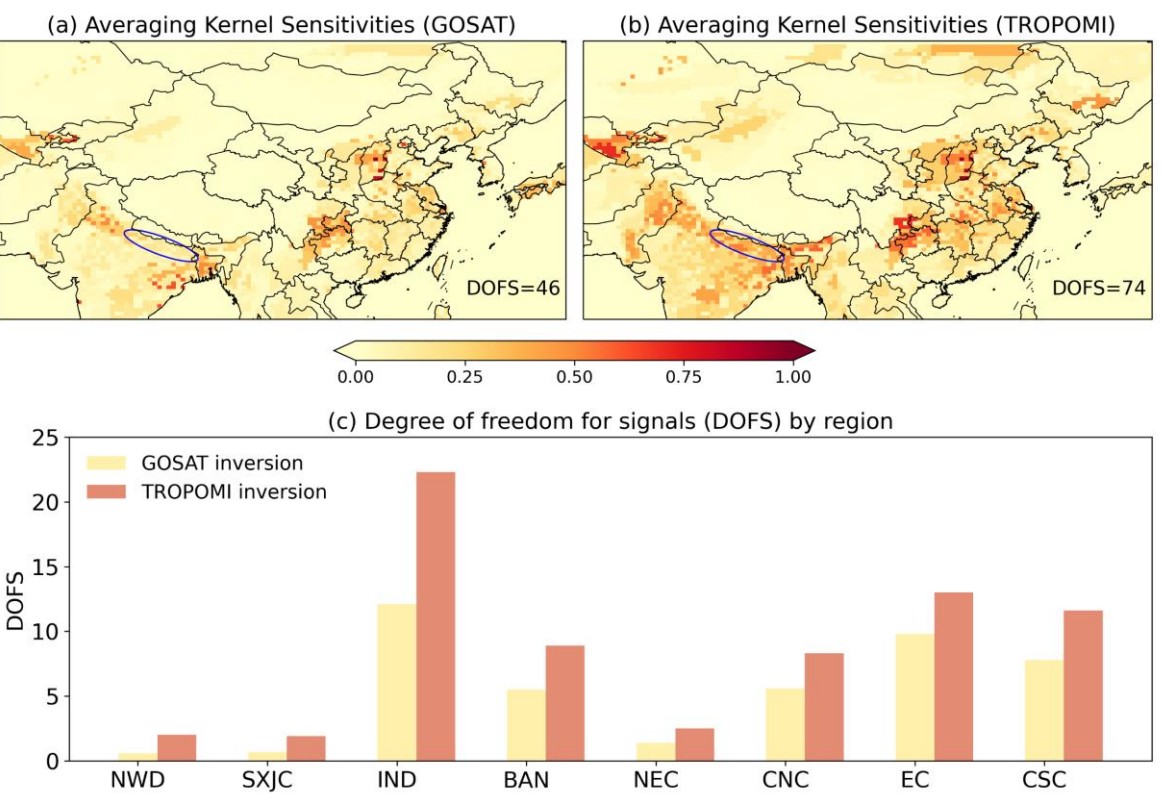

**Figure 7: Averaging kernel sensitivities for GOSAT (a) and TROPOMI (b) inversions. Values represent the ability of observations to constrain methane emissions (0 = not at all, 1 = perfectly). The east Indo-Gangetic Plain is marked by blue rectangles. Panel (c)**
**compares the DOFS of regional emissions constrained by TROPOMI and GOSAT inversions.**

Figure 7 compares the ability of TROPOMI and GOSAT inversions to constrain the distribution of methane emissions, measured by averaging kernel sensitivities (diagonal elements of the averaging kernel matrix). The sum of averaging kernel sensitivities over a region represents the number of pieces of independent information (also known as degree of freedom for

signals, DOFS) constrained by an observation system. Figure 7 shows that the TROPOMI inversion has a larger DOFS value (74) than does the GOSAT inversion (46) over the East Asia domain. A large difference in DOFS between the two inversions is found in IND (TROPOMI: 23 vs. GOSAT: 13), indicating a weak observational constraint on emissions from IND by the GOSAT inversion, even though it infers a large emission correction.

We further investigate why this correction is inferred by the GOSAT inversion by examining the contribution of individual observations to the correction. This analysis indicates that the correction is primarily driven by observations in Bangladesh (Figure 8a). Low $XCH_4$ biases are found over Bangladesh when we compare the prior simulation to either GOSAT or TROPOMI observations (Figure 4). In the absence of GOSAT observations over the Indo-Gangetic Plain, the inversion partly attributes these $XCH_4$ biases to emissions from the Indo-Gangetic Plain, which is upwind of Bangladesh most of the time,

leading to a substantial upward correction of emissions from IND. In contrast, the $XCH_4$ bias over Bangladesh is corrected locally by the TROPOMI inversion. In this case, only small corrections are inferred for emissions from the Indo-Gangetic Plain and the corrections are informed mainly by observations over the Indo-Gangetic Plain (Figure 8b).

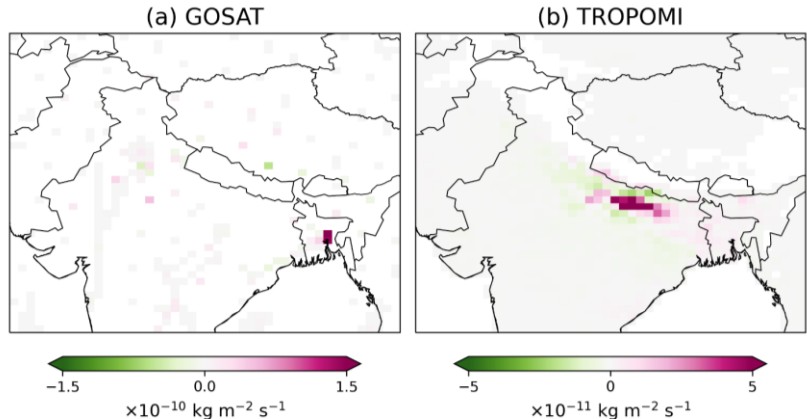

**Figure 8: Contribution of individual observations to the correction of emissions from the Indo-Gangetic Plain by the (a) GOSAT**
**and (b) TROPOMI inversions. This is done by decomposing the computation of Eq. (4). Results are aggregated on the inversion grid. The scales are different between the two panels.**

### 4.3.3 Regional boundary conditions

Our evaluation against surface observations shows improved agreement at background sites (i.e., PDI, UUM, and WLG) by both inversions (Table 1). This is achieved through simultaneous optimization for biases in boundary conditions together with emissions. As WLG, UUM, and PDI are respectively sensitive to the west, north, and south boundaries, this result suggests that satellite observations can correct biases along these boundaries, supporting our inversion configuration. Furthermore, we find that a sensitivity inversion not optimizing for boundary condition biases (S0) cannot reduce large prior biases at WLG and PDI and leads to unrealistically high methane emissions over East Asia (222 Tg a$^{-1}$) including China (102 Tg a$^{-1}$).

An exception in Table 1 is LLN (a high-mountain background site in the southeast of the domain) where biases are increased by both inversions. Although the site AMY is also close to the east boundary, it has little influence from the southeast monsoon (Figure 5c). The biases show strong seasonality, with the largest occurring in summer consistent with ocean-to-land (southeast to northwest) transport by summer monsoon. Our analysis suggests that this increase in biases is caused by large adjustments at the east boundary (GOSAT: 4.5 ppbv; TROPOMI: 25.4 ppbv) rather than changes in methane emissions (Figure 5). This result indicates that satellite observations that are mainly over land are insufficient to constrain the east boundary which consists mainly of ocean.

We then assess the impact of biases along the east boundary on inferred methane emissions. We perform sensitivity inversions using varied levels of fixed (not optimized by the inversion) east boundary conditions, and find relatively small effects on quantifying annual emissions as expected from prevailing westerlies in midlatitudes. A positive bias of 10 ppbv would result in a reduction of annual methane emissions by 3.3 Tg a$^{-1}$ (~2%) over the East Asia domain, 1.8 Tg a$^{-1}$ (~2%) over China, and 0.75 Tg a$^{-1}$ (~3%) over EC (the most affected region) (Figure 9). Although the inversion has a weak constraint on the east boundary conditions, it does not have a great influence on the posterior emissions.

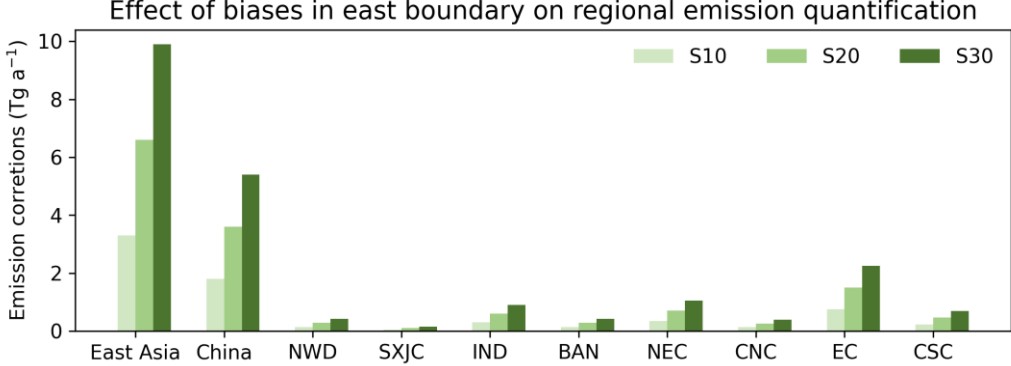

**Figure 9: Impact of biases in the east boundary on quantification of annual methane emissions. Inversions are performed by using fixed east boundary conditions. Sensitivity results are computed from perturbing these fixed east boundary conditions by 10 (S10), 20 (S20), and 30 (S30) ppbv.**

**5 Conclusions**

We estimate methane emissions from East Asia for 2019 by applying atmospheric methane column retrievals from two different satellite instruments (GOSAT and TROPOMI) to a high-resolution regional inversion framework, in which methane emissions are optimized on 600 spatial clusters with up to about half degree horizontal resolution. Our objective is to assess if consistent methane emissions from East Asia are inferred from inversion of GOSAT and TROPOMI observations. This information adds to the uncertainty characterization of satellite-data-based methane emission quantification.


The two inversions estimate a consistent magnitude of methane emissions from East Asia (TROPOMI: 142.7 Tg a$^{-1}$; GOSAT: 142.6 Tg a$^{-1}$) as compared to prior estimate (130 Tg a$^{-1}$) but differ by ~10% in China (TROPOMI: 73.7 Tg a$^{-1}$; GOSAT: 66.4 Tg a$^{-1}$). Comparisons at the regional scale show that the GOSAT and TROPOMI inversions find consistent results over Central North China, Central South China, Northeast China, and Bangladesh, where the inferred emissions differ by less than 2.6 Tg

a$^{-1}$. However, the two inversions show large differences over some of the important regions including northern India and East China. The inferred methane emissions by GOSAT observations are 7.7 Tg a$^{-1}$ higher than those by TROPOMI over northern India but 6.4 Tg a$^{-1}$ lower over East China. Large differences in inferred emissions are also found in northwestern China and Kazakhstan (SXJC and NWD). These findings from the comparison of the GOSAT and TROPOMI inversions are robust against varied inversion configurations.


We evaluate the inversion results by comparing GOSAT and TROPOMI posterior simulations with independent observations. We find that independent ground-based *in situ* observations at AMY and total column observations at XH and HF are more compatible with lower methane emissions from East China inferred by the GOSAT inversion than those by the TROPOMI inversion. We also indirectly evaluate against tropospheric aircraft observations over India during 2012–2014 by using a

consistent GOSAT inversion of earlier years as an inter-comparison platform, which favors lower methane emissions from northern India inferred by the TROPOMI inversion over those by the GOSAT inversion.

The fact that high East China emissions inferred from TROPOMI are inconsistent with independent observations suggests high regional biases in TROPOMI retrievals over East China. Large retrieval differences between GOSAT and TROPOMI are also

found in the northwestern China and Kazakhstan, which also leads to substantially higher methane emissions inferred by the TROPOMI inversion. Unfortunately, we do not have independent observations to evaluate the results in these two regions. However, we note that large TROPOMI XCH$_4$ variations in Kazakhstan and northern Xinjiang are coincident with seasonal changes in surface albedo, suggesting possibly over-correction of surface albedo dependent biases in TROPOMI retrievals at the regional level.


The two inversions show large discrepancies in emissions over northern India along the Indo-Gangetic Plain, although GOSAT and TROPOMI XCH$_4$ values agree reasonably well. We find that the discrepancy in emissions from the Indo-Gangetic Plain is related to differences in data coverage. In the absence of GOSAT observations over the Indo-Gangetic Plain, the inversion attributes the model-observation differences in XCH$_4$ over Bangladesh partly to its upwind region. In contrast, the TROPOMI
inversion finds little emission correction based on the observations over the Indo-Gangetic Plain and attributes the XCH$_4$ differences over Bangladesh primarily to local emissions.

Both inversions show improved agreement at background sites supporting our optimization of boundary condition biases. An exception is LLN where both inversions show large positive concentration biases against *in situ* measurements, which results
from over-corrections at the eastern boundary by inversions. However, our simulations demonstrate that methane concentration biases at the eastern boundary have relatively small impacts on annual emission inferences. The newer version of the TROPOMI methane product includes glint-mode ocean observations, which may improve the optimization of eastern boundary conditions.

## Appendix A. Approximation to the inverse of the error correlation matrix

The observational error covariance matrix is decomposed as $\mathbf{S_0} = \mathbf{\Sigma C \Sigma}$, where $\mathbf{\Sigma}$ is a diagonal matrix while $\mathbf{C}$ is in general non-diagonal. The inverse of $\mathbf{S_0}$ can then be written as $\mathbf{S_0^{-1}} = \mathbf{\Sigma^{-1} C^{-1} \Sigma^{-1}}$. However, the computations of $\mathbf{C^{-1}}$ and $\mathbf{S_0^{-1}}$ quickly become intractable as the dimension of the $\mathbf{C}$ matrix ($m$) grows. We therefore seek for $\mathbf{\tilde{C}^{-1}}$ that approximates $\mathbf{C^{-1}}$ but is easy to compute. To do so, we assume that $\mathbf{\tilde{C}^{-1}}$ is a diagonal matrix.

For clarity, we denote $\mathbf{C^{-1}}$ as $\mathbf{X}$ and $\mathbf{\tilde{C}^{-1}}$ as $\mathbf{\tilde{X}}$. We have the following linear system by definition:

$$\mathbf{CX} = \mathbf{I}, \tag{A.1}$$

where $\mathbf{I}$ is an identity matrix. To find $\mathbf{X}$ is to find its column vectors $\boldsymbol{x}_i$ such that

$$\mathbf{C}\boldsymbol{x}_i = \boldsymbol{e}_i, \quad i = 1,2,\cdots,m. \tag{A.2}$$

Here $\boldsymbol{e}_i = (0,\cdots,1,\cdots,0)^T$ is a unit vector with its $i$th element being 1 and the rest 0.

By assuming that $\mathbf{\tilde{X}}$ is diagonal, we impose the condition that its column vectors $\tilde{\boldsymbol{x}}_i \in \text{span}\{\boldsymbol{e}_i\}$. We apply the oblique projection technique to find the solution for $\tilde{\boldsymbol{x}}_i$, such that the residual vector $\boldsymbol{e}_i - \mathbf{C}\tilde{\boldsymbol{x}}_i$ is orthogonal to the 1-dimension subspace spanned by $\mathbf{C}\boldsymbol{e}_i$ (Saad, 2003). Hence, we have

$$(\mathbf{C}\boldsymbol{e}_i)^T (\boldsymbol{e}_i - \mathbf{C}\tilde{\boldsymbol{x}}_i) = 0. \tag{A.2}$$

Solving the equation yields

$$\tilde{x}_{ii} = \frac{C_{ii}}{||\mathbf{C}\boldsymbol{e}_i||_2^2} = \frac{1}{\sum_{j=1}^{m} C_{ij}^2}, \tag{A.3}$$

where $\tilde{x}_{ii}$ is the $i$th element of $\tilde{\boldsymbol{x}}_i$ and $|| \cdot ||_2$ represents the L-2 norm. Because $\mathbf{C}$ is a correlation matrix, its diagonal element $C_{ii}$ is equal to 1.

Consequently, we obtain

$$\mathbf{\tilde{C}^{-1}} = \mathbf{\tilde{X}} = \text{diag}\left(\frac{1}{\sum_{j=1}^{m} C_{1j}^2}, \frac{1}{\sum_{j=1}^{m} C_{2j}^2}, \cdots, \frac{1}{\sum_{j=1}^{m} C_{mj}^2}\right). \tag{A.4}$$

Note that computation of $\mathbf{\tilde{C}^{-1}}$ can be readily parallelized for speed-up.

The diagonal elements of $\mathbf{\tilde{C}^{-1}}$ can be interpreted as the weight for individual observations. The weight is 1 for an independent observation $i$ uncorrelated with any other observations ($C_{ii} = 1$ and $C_{ij} = 0$ for $i \neq j$), while the weight can be substantially smaller than 1 for an observation with strong correlation with others (many non-zero $C_{ij}$ terms or large $C_{ij}$ terms for $i \neq j$).

## Data availability

The TROPOMI methane observations are from https://ftp.sron.nl/open-access-data-2/TROPOMI/tropomi/ch4/14_14_Lorente_et_al_2020_AMTD (last access: 29 December 2021). The GOSAT methane observations are the University of Leicester GOSAT Proxy $XCH_4$ v9.0 (ceda.ac.uk), accessible through https://data.ceda.ac.uk/neodc/gosat/data/ch4/nceov1.0/CH4_GOS_OCPR/, or the Copernicus Climate Data Store (https://cds.climate.copernicus.eu/). Surface observations at PDI are downloaded from https://gaw.kishou.go.jp/. Surface

observations at AMY, LLN, UUM, and WLG and aircraft observations from the CARIBIC project are available via the NOAA ObsPack $CH_4$ product (https://gml.noaa.gov/ccgg/obspack/index.html). The Xianghe FTIR $CH_4$ data are accessible through https://doi.org/10.18758/71021049. The Hefei FTIR $CH_4$ from TCCON network can be accessed by contacting Prof. Cheng Liu at University of Science and Technology of China.

## Author Contributions

RL and YZ designed the study. RL performed the inverse modelling with contributions from YZ, WC, PZ, JL, ZQ, and ZC. RL analysed and interpreted results with contributions from YZ, CC, HM, GS, ZQ, MZ, RJP, HB, AL, JDM, and IA. RJP and HB provided the GOSAT methane retrievals. AL, JDM, and IA provided the TROPOMI methane retrievals. MZ and PW provided ground based FTIR methane retrievals at the Xianghe site. RL and YZ wrote the paper with inputs from all authors.

## Acknowledgements

This research is supported by the National Key Research and Development Program of China (2021YFB3901000) and the National Natural Science Foundation of China (42007198). We thank the team that realized the TROPOMI instrument and its data products, consisting of the partnership between Airbus Defense and Space Netherlands, KNMI, SRON, and TNO, commissioned by NSO and ESA. Sentinel-5 Precursor is part of the EU Copernicus program, Copernicus (modified) Sentinel-5P data (2018-2020) have been used. AL acknowledges funding through the TROPOMI national programme from NSO. We

thank the Japanese Aerospace Exploration Agency, National Institute for Environmental Studies, and the Ministry of Environment for the GOSAT data and their continuous support as part of the Joint Research Agreement. Robert J. Parker and Hartmut Boesch acknowledge support from the UK National Centre for Earth Observation funded by the National Environment Research Council (NE/R016518/1 and NE/N018079/1) and the Copernicus Climate Change Service (C3S2_312a_Lot2). The research of GOSAT retrievals used the ALICE High Performance Computing Facility at the University of Leicester. We thank

the TCCON community and Weidong Nan (IAP) for supporting the Xianghe FTIR measurements, and the Department of Precision Machinery and Precision Instrumentation, University of Science and Technology of China (Cheng Liu's group) for providing ground-based remote sensing data (Hefei) with their own independent intellectual property rights. We thank Korea Meteorological Administration, Viet Nam Meteorological and Hydrological Administration, China Meteorological

Administration, and NOAA for providing surface measurements through GLOBALVIEWplus $CH_4$ ObsPack and WDCGG. We thank the High-performance Computing Center of Westlake University and National Supercomputing Center at Wuxi for facility support and technical assistance.

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
