# Peer review of "East Asian methane emissions inferred from high-resolution inversions of GOSAT and TROPOMI observations: a comparative and evaluative analysis"

_Atmospheric Chemistry and Physics, 2022_

## Author Comment (AC1)

We thank the reviewer for constructive comments that help improve the manuscript. Our responses to the comments are in blue.

The paper by Liang et al. compares regional inverse estimates of methane ($CH_4$) surface fluxes in East-Asia for the year 2019. The driver data are GOSAT and TROPOMI satellite observations of the column-average mole fractions $XCH_4$. Liang et al. describe the methodology based on the GEOS-CHEM transport model and a regularized inversion. They compare inversions of GOSAT and TROPOMI $XCH_4$ and find good agreement for some, substantial discrepancies for other regions. Comparisons to independent data sets serve as guidance to explain the regional differences.

**Scope:** I am a bit puzzled of what the overall goal of the study is. Is it a budget report of East Asian methane emissions or is it an evaluation of GOSAT and TROPOMI biases? The former would require more complete error analyses, more than a year of data, and more extensive discussions of previous work. For the latter, I would argue that the manuscript lacks completeness in terms of discussing error sources (see comment below). I recommend making the overall goal of the study clearer and revising the paper in the view of that goal.

We now clarify our objective in the manuscript (abstract, introduction, and conclusion). We focus on understanding the uncertainty in posterior methane fluxes arising from using different satellite data, which is one of many uncertainty sources for an inverse analysis. We do not intend to provide a comprehensive budget report for East Asia. Our analysis is also more than just evaluation of GOSAT and TROPOMI retrieval biases, as the discrepancy of posterior methane fluxes is related to not only retrieval biases but also other factors such as data coverage. We now state in the text that "*the main objective is to assess the consistency of methane fluxes inferred from the two sets of satellite data that differ in their data coverage and regional accuracy, adding information to the uncertainty characterization of satellite-based methane emission accounting.*". We emphasize that "*the analyses are conducted with identically configured inversions to isolate the effects of observation data*".

**Proxy-CH4:** Generally, the main (and, I believe, conceptually limiting) error source of the proxy method (GOSAT) must be discussed more thoroughly. It is the errors of the $CO_2$ fields that are used to construct $XCH_4$ from the raw $CH_4/CO_2$ ratio. Any (e.g. regionally correlated) errors in the prescribed $CO_2$ fields (typically taken from models) will map into respective errors in $XCH_4$. In fact, others [Schepers et al., JGR, 2012, https://doi.org/10.1029/2012JD017549] have compared proxy and full physics methods in the early days of the GOSAT mission. They found that, in a case study for India, erroneous CarbonTracker $CO_2$ fields caused biases in proxy $XCH_4$ data [Fig. 9 and 10 and related discussion in Schepers et al.]. The paper must examine and discuss this source of error to balance the discussion of scattering induced errors of the full-physics method (TROPOMI). To the best of my knowledge, the current version of the

UoL proxy algorithm uses a model ensemble for $CO_2$-rescaling. One could try to estimate the error by looking at the spread of these (and potentially other) models in the investigated regions.

We add discussion on biases of proxy $XCH_4$ data in IND due to errors in the $CO_2$ field in Section 4.3.1. We cite Schepers et al. (2012) and Parker et al. (2015) in the discussion. We also mention broadly in the introduction section that the proxy method is subject to errors in specified $CO_2$ columns. We have not investigated the spread of $CO_2$ fields used in the UoL product as suggested, as our objective is not to assess error sources of a retrieval product. We aim to assess the impact of different retrievals on the inversed methane fluxes. Both retrieval biases and data coverage affect the inversion results. We show that retrieval biases between GOSAT and TROPOMI are relatively small over India, but they have large differences in data coverage and density.

**Setup of the inverse problem:** I wonder about the setup of the inverse problem. If I get it right, the parameter vector contains 600 spatial elements which represent spatially distributed annual surface fluxes. I find this a mismatch of spatial and temporal scales. While the inversion is free to optimize a lot of spatial detail, any sub-annual temporal variability of fluxes is imposed. Given further, that the measurement vector contains daily $XCH_4$ data, I would argue that the temporal resolution of the inversion is at odds. The authors should discuss this aspect and provide sensitivity studies showing that their choice does not induce biases (e.g. by imposed seasonality).

Further, the authors have chosen to represent the prior covariance in relative terms (50%) with respect to the prior. This choice imposes that the spatial structure of posterior fluxes will be very similar to the one of the prior fluxes (simply because changing a small flux by 50% (or likewise) remains a small flux). This is clearly visible when comparing Fig. 2 and 4 (even though the log-scale in Fig. 2 needs some defiant eyeballing). The authors should clearly state the consequences of this assumption.

We perform additional sensitivity inversions to address the concerns raised by the reviewer. We first conduct a seasonal inversion, which shows similar correction pattern as the annual inversion (including the discrepancies between the GOSAT and TROPOMI inversions over East China and northern India, and the agreement over Northeast China, Central South China, and Bangladesh). The seasonal inversion in general infers smaller corrections than does the annual inversion (given the same error assumptions) because of less observations during a shorter period.

[Figure]

We then conduct an inversion with augmented prior errors of 100%, which again leads to similar correction patterns as the base inversion (agreement in NEC, CSC, BAN and disagreement in EC, IND, NWD, and SXJC). Results of this inversion tend to be noisier and generally have greater magnitudes because of less prior constraints, but the effect is overall small (by ~ 6 Tg a$^{-1}$ over the whole domain).

[Figure]

These sensitivity inversions show that the agreements and discrepancies in posterior methane emissions between GOSAT and TROPOMI inversions are robust against perturbations of inversion setups. These results are now presented in Section 4.1 and supplementary information.

**Inverse method:** Equation 1 is the cost function of the inverse method. It is the classic regularization setup with a prior mismatch and a least squares measurement term where one term is scaled by a regularization parameter which the authors determine according to Figure S3. If I understand correctly, the condition on the selected regularization parameter is that the scaled least-squares term and the prior term impose equal cost. Why would one set such a condition when aiming at evaluating the information content of different data sets? In my understanding, this particular condition implies that whatever your measurement data are (be it dense or sparse, accurate or not), you force the inversion to deliver roughly the same degrees of freedom (for a given prior constraint). Figure 7 appears to confirm this conclusion: while GOSAT and TROPOMI have vastly different data density, the information content of the inversion is roughly the same. In consequence, the presented findings on degrees of freedom would not in any way represent the "natural" information content of the data but they are driven by design of the inverse method.

Generally, I would think that an L-curve method should work better for getting a regularization parameter that actually represents the information content of the data [see the cover (or chapter 4.6) of the book by Per Christian Hansen cited in the manuscript].

We now add more clarification in Section 3.2. The regularization $\gamma$ is necessary here because error correlations are omitted in specified $S_O$ (which is assumed to be diagonal for computational reasons). The extent of error correlations is different for different satellite data because of varied data density.

We have tried using L-curve method to determine a regularization parameter but found no apparent inflection points in the value range $0 \sim 1$. We then determine $\gamma$ following Lu et al. (2021) (section 2.4), which are also used by Qu et al. (2021) and Chen et al. (2022). The theoretical basis of the Lu et al. (2021) method is that if inversion results are consistent with specified errors ($S_A$ and $S_O$), $(x - x_A)^T S_A^{-1}(x - x_A)$ should follow a chi-square distribution with $n$ degrees of freedom and $(y - F(x))S_O^{-1}(y - F(x))$ a chi-square distribution with $m$ degrees of freedom.

**Discussion:** The posterior error bars of the satellite inversions (e.g. line 226f) are very small. I assume that they only represent the propagated measurement errors according to equation (4) (and line 185f) and that model transport errors, representativeness errors, more systematic measurement errors are neglected. When comparing the satellite-derived emissions to other studies (line 230ff), the reported error bars should be representative of the full error budget.

Our purpose here is not to compare with results from other studies or report a methane budget with comprehensively quantified uncertainties. We aim to assess the impact of different satellite observations on inversed methane fluxes. As we use identical inversion setups for the two inversions, systematic error sources mentioned by the reviewer would have similar effects on the two inversions, and therefore do not affect their comparisons. Errors derived from Eq. (4) is what we need to determine whether the difference in Fig. 3 is statistically significant, as it expresses random errors of the posterior estimates. We now clarify in the caption of Fig. 3.

**References**

Chen, Z., Jacob, D. J., Nesser, H., Sulprizio, M. P., Lorente, A., Varon, D. J., Lu, X., Shen, L., Qu, Z., Penn, E., and Yu, X.: Methane emissions from China: a high-resolution inversion of TROPOMI satellite observations, Atmos. Chem. Phys., 22, 10809-10826, https://doi.org/10.5194/acp-22-10809-2022, 2022.

Lu, X., Jacob, D. J., Zhang, Y., Maasakkers, J. D., Sulprizio, M. P., Shen, L., Qu, Z., Scarpelli, T. R., Nesser, H., Yantosca, R. M., Sheng, J., Andrews, A., Parker, R. J., Boesch, H., Bloom, A. A., and Ma, S.: Global methane budget and trend, 2010–2017: complementary of inverse analyses using in situ

(GLOBALVIEWplus CH4 ObsPack) and satellite (GOSAT) observations, Atmos. Chem. Phys., 21, 4637-4657, https://doi.org/10.5194/acp-21-4637-2021, 2021.

Parker, R. J., Boesch, H., Byckling, K., Webb, A. J., Palmer, P. I., Feng, L., Bergamaschi, P., Chevallier, F., Notholt, J., Deutscher, N., Warneke, T., Hase, F., Sussmann, R., Kawakami, S., Kivi, R., Griffith, D. W. T., and Velazco, V.: Assessing 5 years of GOSAT Proxy $XCH_4$ data and associated uncertainties, Atmos. Meas. Tech., 8, 4785-4801, https://doi.org/10.5194/amt-8-4785-2015, 2015.

Qu, Z., Jacob, D. J., Shen, L., Lu, X., Zhang, Y., Scarpelli, T. R., Nesser, H., Sulprizio, M. P., Maasakkers, J. D., Bloom, A. A., Worden, J. R., Parker, R. J., and Delgado, A. L.: Global distribution of methane emissions: a comparative inverse analysis of observations from the TROPOMI and GOSAT satellite instruments, Atmos. Chem. Phys., 21, 14159-14175, https://doi.org/10.5194/acp-21-14159-2021, 2021.

Schepers, D., Guerlet, S., Butz, A., Landgraf, J., Frankenberg, C., Hasekamp, O., Blavier, J.-F., Deutscher, N. M., Griffith, D. W. T., Hase, F., Kyro, E., Morino, I., Sherlock, V., Sussmann, R., and Aben, I.: Methane retrievals from Greenhouse Gases Observing Satellite (GOSAT) shortwave infrared measurements: Performance comparison of proxy and physics retrieval algorithms, J. Geophys. Res. Atmos., 117, https://doi.org/10.1029/2012JD017549, 2012.

---

## Author Comment (AC2)

We thank the reviewer for constructive comments that help improve the manuscript. Our responses to the comments are in blue.

Inverse methane emissions over East Asia are estimated and compared in this research for the year 2019 using data from two satellite observations, GOSAT and TROPOMI. Based on the GEOS-Chem transport model and analytical Bayesian inversion, Liang et al. developed the regional inversion framework. Comparisons at the regional level reveal consistency overall, but significant variation in some areas. The authors analyze the observations and independent measurements in further detail and argue that the variations can be explained by the data coverage and various retrieval techniques. Some arguments, however, need more analysis or are not convincing enough. Only after the authors address the following concerns can I recommend the paper to be published.

**General comments**

1, The overall goal of this paper is not very clear. Several key points are mixed up in this manuscript, but the authors do not break it down into individual points for this study. If the authors intend to assess the impact of the various satellite retrievals, I would suggest using one satellite but two retrieval products (for GOSAT: "proxy" v.s. "full-physics"; TROPOMI: official dataset v.s. WMFD). If the authors want to show the robustness of the inversion system they built, I suggest discussing error sources in detail and showing intermitted results.

We now clarify our objective in the manuscript (abstract, introduction, and conclusion). We now state in the text that "*the main objective is to assess the consistency of methane fluxes inferred from the two sets of satellite data that differ in their data coverage and regional accuracy, adding information to the uncertainty characterization of satellite-based methane emission accounting.*". Observations from different satellites are being applied in varied inversion studies to quantify regional emissions, but it is yet unclear if or to what extent different retrievals result in consistent methane emission estimates. It is often difficult to isolate the effect based on these studies because of confounding effects from differences in inversion configurations. We conduct the analyses with "*identically configured inversions to isolate the effects of observation data*".

We use the GOSAT UoL proxy retrieval and the TROPOMI full-physics retrieval in our evaluation. The assessment of inversion using these products are relevant because they have been widely used in inversion studies on regional and global scales. They are sufficiently different (different instruments, retrieval algorithms, data coverage, and regional accuracy), so our results bracket the uncertainty that may arise from using different satellite observation data. Evaluation of different retrievals from one

satellite, as suggested by the reviewer, is also useful, but that will be a completely different study focusing more on retrieval algorithms.

To avoid misinterpretation, we now emphasize that "*there are other operational and science retrieval products available from both GOSAT and TROPOMI measurements*" and "*our analyses and conclusions are specific to the retrieval products used here*".

2, In both the abstract and discussion, the authors mention the large discrepancy over certain areas in East Asia is caused by retrievals. The cost imposed by the least-squares term and the prior term, however, appears to be equal. Thus, the changes in posterior emissions are mainly driven by the inverse system but not observations.

As expected from a Bayesian inversion, posterior emission estimates are a combination of observations and prior estimates and are affected by the inversion configurations. We isolate the effects of observation data on posterior emissions by applying different satellite data to identically configured inversions. We now clarify this point in various places throughout the manuscript. We also add results from sensitivity inversions with perturbed inversion configurations, which shows that the difference between results of the two inversions is robust against these perturbations. This is now shown in Section 4.1 and supplementary information.

3, The comparisons between simulations (with a priori and a posterior) and other independent measurements also indicate that increasing the emission intensity is ineffective to improve the result (see specific comment 6) in background areas. Does it imply that the model is unable to adequately capture the variations in these areas or that certain sources are missing from the same grid cell? The common problem in inverse modeling is the missing sources in a priori emission inventory. Again, if the authors aim at evaluating the emissions in China, please add more discussion on this aspect.

Table 1 shows that the inversion reduces biases substantially at background sites at WLG and PDI, which indicates better performance in the background area. The inversions are unable to improve performance for temporal variability ($R^2$) at the surface sites. This is largely due to the fact that optimization of methane emissions is done on an annual basis and other factors include model transport errors and observation representativeness. We now clarify in a footnote of Table 1.

**Specific comments**

1, Line 50-55: Please add more information about other methods to derive methane emissions as well as other satellites that are currently in service for methane monitoring (Sentinel-2, GHG-sat,

etc.). The introduction here can be more comprehensive.

Sentinel-2 and GHGSat are point-source imagers that are not suitable for global and regional emission quantification. We describe and clarify in the introduction section that we are interested in area flux mappers such as GOSAT and TROPOMI.

2, Line 95-100: As far as I know, the new version of TROPOMI has already been reprocessed. And they provide the data over the ocean (glint-mode). If the authors downloaded the official operational product, I strongly suggest using the reprocessing data. The operational product comes in a variety of versions, each of which contains various errors and biases that might cause inconsistency in error analysis. Please check/specify if the data in 2019 comes from the same version.

Additionally, retrieving data over the ocean (typically retrieved under sun-glint conditions) differs from retrieving data over land. It might cause discontinuity from land to ocean. Do authors check if there are any corresponding biases in GOSAT data?

All data in 2019 comes from the SRON S5P-RemoTeC scientific TROPOMI CH$_4$ dataset, version 14 (Lorente et al., 2021). We downloaded data from an ftp site: https://ftp.sron.nl/open-ccess-data-/TROPOMI/tropomi/ch4/14_14_Lorente_et_al_2020_AMTD/.

According to the SRON RemoTeC-S5P scientific XCH4 data product Product User Guide, the algorithm in the scientific product by Lorente et al. (2021) is implemented later in the official operational product of version 2.02.00. However, the published operational data are only available from July 2021 to November 2021, which does not cover the study period. The more recent TROPOMI data v 2.03.01 (over land and ocean) are only available from November 2021 to July 2022. The 2019 official operational product has not been reprocessed so far with the new algorithm. (Page 5 of S5P-MPC-SRON-PRF-CH4_v2.4.1_2.2_20220720 (copernicus.eu))

GOSAT glint-mode data are continuous from land to ocean, as shown below (presented on the 2° × 2.5° grid). A similar conclusion is reached by Qu et al. (2022) who used GOSAT observations over both land and ocean for a global study.

[Figure]

GOSAT

3, The diverging colormap of Figure 2 causes confusion. It is better to use a monotonically increasing colormap.

The figure has been updated following the suggestion.

4, Line 160. The anthropogenic emission from EDGAR v4.3.2 is relatively out of date, and which has also been found that the emissions have been overestimated in many areas. Why do authors not use the later version (latest: EDGAR v6)? At least, the authors should mention/estimate known biases in EDGAR v4.3.2.

We find the difference between different versions of EDGAR is insignificant in terms of the emission magnitude and spatial distribution over East Asia. We do not expect this newer bottom-up inventory will change the findings and conclusion of our study. We now present this comparison in Section 3.1 and a supplementary figure (see also below).

[Figure]

[Figure]

5, About Figure 4, either in GOSAT or TROPOMI inversion, the spatial differences show a strong spatial correlation between a priori and a posterior (a v.s. c and b v.s. d). Is it caused by the assumption of a priori covariance?

It is unclear to us what the reviewer meant here. The differences between the prior simulation and GOSAT observations are well corrected. The differences between the prior simulation and TROPOMI observations are also reduced but not fully corrected, reflecting constraints from prior estimations.

6, About Table 1, there are no improvements in the values of $R^2$. The low $R^2$ may imply the model lack repetitiveness in some places (considering they are background stations). Additionally, after being constrained by satellite measurements, the negative biases with the a priori inventory simply turn to positive biases, demonstrating that adjusting the emission intensity does not improve the outcomes of simulations.

The objective of Table 1 is to evaluate the discrepancies from the two inversions with independent observations, which is mainly shown by data at AMY, XH, and HF. Nevertheless, Table 1 does show substantial reduction in biases in background sites WLG and PDI by both inversions. The large negative bias in the prior simulation at WLG (the background site upwind East China where XH and HF are located) also indicates that small prior biases for XH and HF are due to the wrong reason (underestimated background concentrations). We now add clarification along with Table 1. We discuss in Section 4.3.3 reasons for poor performance at LLN.

Many factors may limit improvement of $R^2$ for independent observations at a single station. The main reason is that this inversion also does not seek to optimize temporal variations in emissions. Other factors include model transport errors and observation representativeness. We now clarify in Table 1.

7, Line 310, section 4.3.1. Figure 3(a) show higher emission corrections than (b) while Figure 6(a) displays a small variation in concentration in IND. However, the variation of XCH$_4$ in EC demonstrates the consistency of Figures 3 and 6. Any explanations for this?

This is extensively discussed in Section 4.3.2. Differences in data coverage are also an important factor for the discrepancy in inferred emissions.

8, section 4.3.1. How do the sampling biases in different seasons and regions affect the comparisons between GOSAT and TROPOMI?

Figure 6b shows regional biases as a function of seasons. Spatial distributions of the comparison are also shown by season below. There are almost no coincident GOSAT and TROPOMI observations in Southeast Asia, South India, and Bangladesh during July and September. We assess the impact by performing additional inversions that optimizes emissions by season instead of annually. We find similar correction patterns as our main results. This result is now presented in Section 4.1 and in supplementary figures.

[Figure]

9, Line 390, Section 4.3.3. The authors argue that the lack of observations over the ocean leads to unrealistic enhancement of XCH$_4$. However, there are no sources over the ocean, the strong

enhancement at the southeast corner is more likely caused by the model's erroneous processes of the transport. For example, the boundary condition of the regional model is updated by the output of the global model, which may contain bugs/errors.

The ocean data are useful for the inversion to reduce biases in the boundary conditions given by the global model. This mechanism is included in our inversion and is described in Section 3.2.

**Reference**

Lorente, A., Borsdorff, T., Butz, A., Hasekamp, O. P., aan de Brugh, J., Schneider, A., Wu, L., Hase, F., Kivi, R., Wunch, D., Pollard, D. F., Shiomi, K., Deutscher, N. M., Velazco, V. A., Roehl, C. M., Wennberg, P. O., Warneke, T., and Landgraf, J.: Methane retrieved from TROPOMI: improvement of the data product and validation of the first 2 years of measurements, Atmos. Meas. Tech., 14, 665-684, https://doi.org/10.5194/amt-14-665-2021, 2021.

Qu, Z., Jacob, D. J., Zhang, Y., Shen, L., Varon, D. J., Lu, X., Scarpelli, T., Bloom, A., Worden, J., and Parker, R. J.: Attribution of the 2020 surge in atmospheric methane by inverse analysis of GOSAT observations, Environmental Research Letters, 17, 094003, https://doi.org/10.1088/1748-9326/ac8754, 2022.

---

## Referee Report (RR1)

It was my pleasure to review the manuscript. I understand that my comments are coming at a relatively late stage in the review process, which is reflected in the generally high quality of the manuscript at this point. I suspect that many of the initial problems have been worked out by now, and some interesting, additional findings have emerged along the way. In its present state, I would consider this manuscript appropriate for publication with only minor adjustments, as outlined below.

One of the most scientifically interesting findings is the temporal dependence in the differences between the GOSAT and the TROPOMI data (compared through a model), shown in Figure 6. The fact that the largest seasonal features correlate with changes of surface albedo suggests directions for the future improvement of (TROPOMI) retrievals algorithms. I think it would be worth highlighting this finding in the abstract as well.

Like the last reviewer, I am also puzzled by how small the error bars on the posterior flux estimates are. The authors mention that these uncertainties are "optimistic", as they neglect systematic uncertainties due to the inversion setup. This is not ideal, and should be improved in the future, but does not invalidate the focus of the GOSAT-TROPOMI comparison presented here.

**Minor comments that need to be addressed:**

L326: The WLG site show(s) a relatively low posterior what? (Also, it's not the site itself that is unable to capture variability, as this sentence implies.) How can the simulations simultaneously show "relatively good agreement" and "relatively low posterior (correlation)" at WLG? Relative to what? Is it really "day-to-day" variability that is poorly captured in the case of flask data? I guess the sampling is roughly weekly, although the bizarre x-axis in S9 makes this impossible to say for sure (see comment about the supplement below).

Figure 5 is lacking a colour bar!

L506-507: Based on the error bars, only CSC and NEC show consistent (and not just qualitatively similar) posterior emissions. This is important. Either the uncertainties in the posterior emissions estimates are grossly underestimated (likely true), or the datasets are fundamentally inconsistent with one another (also likely true).

Figure S9: Something very strange is going on with the x-axes. It looks like the observations are spaced equally, and the ticks for the months are adjusted to fit. This happens in such a way that there are always measurements on the first of each month, which is suspicious. This is not a normal way of presenting timeseries data, and should be fixed. Are there really only 23 FTIR measurements from HF for the whole year?

**Technical/stylistic comments/typos:**

L30: Suggested rephrasing: The methane emissions inferred from GOSAT observations are … higher than those from TROPOMI observations…

L31: These -> The

L38: I think the "Ganges Plain" is usually referred to as the "Indo-Gangetic Plain". I would recommend changing it throughout.

L67: a number is missing regarding the TROPOMI footprint.

L79: A bit confused by the use of "variable" here. Is it required?

L94: Is it already clear before this study that the two sets of satellite data differ in their "regional accuracy"? If so, a citation is needed! Otherwise, remove this.

L97, but also in abstract and throughout the study: I guess North(ern) India should be capitalized throughout? As is every region of China… This is certainly how Wikipedia does it: https://en.wikipedia.org/wiki/North_India

L114: are -> is

L126: annual average XCH4 on the 0.625°× 0.5° grid for GOSAT and TROPOMI -> XCH4 measured by GOSAT and TROPOMI, annually averaged on the 0.625°× 0.5° grid.

L127: over THE Mongolian…

L129: of multiple measurements fall in -> when multiple measurements fall within

L149: Were the in situ data temporally filtered? Generally nocturnal measurements are not well represented by models.

L163: located distant -> far

L182: more discussed -> discussed further

L183: early -> earlier (or previous)

L190-191: WetCHARTs is an ensemble product, which specific ensemble member was used? Is there any concern about double-counting rice emissions as wetlands?

L204: either "a biased boundary condition" or "biased boundary conditions"

L264: overestimate -> overestimates

L296: estimate -> estimates

L303: applying A traditional regularization

L431: are in India -> either "that are in India" or "in India"

L431: subscript in XCO2

I would really like to see a figure of the 600 spatial clusters included in the supplement.

L540: benefit -> improve

Caption of Figure S10, last line: the bias correction factor greater than -> the bias correction factor is greater than

---

## Author Response (AR2)

**Editor**
*Atmospheric Chemistry and Physics*

25 March 2023

Dear Dr. Stiller,

Thank you for handing our manuscript. In this submission, we include substantial revisions to address the reviewer's concern. Below I summarize the main points and a point-wise response is also included.

First, we add a series of sensitivity inversions to test the robustness of our main findings. There are in total 4 sensitivity inversions configured differently from the main inversion, including (1) a seasonal inversion to address the reviewer's concern on "sub-annual variability" and (2) an inversion whose prior errors are specified for greater flexibility to adjust at locations with small prior methane emissions, which is to address the reviewer's concern on "strongly constrained sub-regional patterns". These sensitivity tests all find large discrepancies between the GOSAT and TROPOMI inversions in East China and northern India, indicating that our main finding is robust. The results are presented in Section 4.1 and Figure S7 and S8. Results of individual sensitivity inversions is also discussed wherever relevant.

Second, we develop and implement a new method to specify the $\mathbf{S_O}$ matrix, which does not involve the regularization parameter $\gamma$ used previously. This is to address the reviewer's concern that the empirical procedure to choose the $\gamma$ value leads to "the total information content for the whole region to be roughly the same". This new method should provide better interpretability than the traditional $\gamma$ method. The method is described in the new Section 3.3 and its mathematical derivation in the Appendix A. Previous inversion result is included into the set of sensitivity tests.

Third, we investigate the reason for the discrepancy found over the Northern India in detail. We add more details to our explanation that this discrepancy is related to differences in data coverage between GOSAT and TROPOMI. Our additional analyses reveal that the discrepancy is also related to the large model-observation mismatch in the downwind Bangladesh region. In the absence of observations over the Gangetic Plain, the GOSAT inversion attributes this mismatch partly to northern India, while the TROPOMI inversion attributes it entirely to local Bangladesh emissions. This result is now added in Section 4.3.2.

Finally, we add more discussion on the retrieval errors, particularly for the $CO_2$ proxy retrieval (Figure S1, Section 2.1, and Section 4.3.1).

We hope that our revised manuscript will be suitable for publication in *Atmospheric Chemistry and Physics*. Thank you for your consideration!

Sincerely,

Yuzhong Zhang
zhangyuzhong@westlake.edu.cn
Westlake University
Hangzhou, Zhejiang, China

We thank the reviewer for constructive comments that help improve the manuscript. Below are our point-to-point responses shown in blue.

(1) without assessing the full error budget:

The discussion of the retrieval errors, in particular the imposed CO2-rescaling of proxy-CH4, remains short and qualitative (despite the comment raised in round 1). In their reply, the authors write that they do not assess these retrieval errors essentially because it is out-of-scope (although there is a section "4.3.1 Regional retrieval bias"). Since (parts of) the differences between the GOSAT and TROPOMI-inversions could originate from regional biases in the respective satellite data and not just differences in data density, I would argue that it is important to make an attempt to quantify such biases. For GOSAT proxy-CH4, this could be as easy as looking at the spread of models that go into the CO2-rescaling of the UoL algorithm (as suggested in round 1). For TROPOMI CH4, looking at the size of the (albedo-driven) bias correction could inform on which regions are more difficult than others.

We now show the modeled $XCO_2$ used by the UoL algorithm and their ranges from the three models in Figure S1. We add in Section 2.1 and Section 4.3.1 discussion on the uncertainty of the $CO_2$ proxy retrieval due to the specified $XCO_2$ field. We have shown the TROPOMI bias correction in Figure S2 and discussed its impact on the inversion in Section 4.3.1 and Figure S11 and S12.

As pointed out by the reviewer, the above information provides some hints about which regions are likely more difficult for retrievals. However, large discrepancy in emission estimation may emerge in regions that do not stand out in this type of retrieval error analysis, for example, East China (because there may be error sources that have not yet been identified). Our approach here is to empirically evaluate the retrieval and the corresponding emission estimates against independent observations, using the transport model as a platform for the comparison (Section 4.2).

For better clarity, we now modify the title of Section 4.3.1 to "Regional differences in $XCH_4$ retrievals", to make it clear that our focus is to explain the differences of inversion results rather than analyzing the sources of errors/biases for different retrievals.

(2) with imposing sub-annual variability:

In response to the comment on "any sub-annual temporal variability of fluxes" being imposed, the authors conducted a sensitivity run where they optimized for seasonal fluxes instead of the annual total. The reply contains a figure (without much explanations) that shows that the results for the seasonal and annual inversions differ substantially in magnitude and for some regions in the sign of the fluxes. Assuming that the figures show the annual fluxes for both inversions – which is the only thing that makes sense – I would argue that attributing the differences to "less observations" is not valid and that this needs further investigations.

We now clarify in the caption that the figure shows annual-averaged emissions from the

seasonal inversion (Figure S7). We remove the statement that the smaller correction inferred from the seasonal inversion is due to less observations.

We now present the results from a series of sensitivity inversions including this seasonal inversion (Section 4.1, Figure S7 and S8). We show that our main findings on the regional emissions are robust against varied inversion configurations. Despite some differences in the posterior solutions, all these sensitivity inversions including this seasonal inversion find that the large discrepancy between the GOSAT and TROPOMI inversions occurs in East China and northern India, which is consistent with the main inversion.

We also examine whether the seasonal inversion better captures the temporal variability in observations from surface sites than does the annual inversion (Section 4.2 and Figure S9). We find the improvement is overall small.

(3) with strongly constraining the sub-regional patterns through a relative prior covariance matrix:
In response to the comment on using a "prior covariance in relative terms" (i.e. define as a percentag of the fluxes), the authors did a sensitivity run where they scaled the prior covariance to represent 100% instead of 50% flux errors. Obviously, this does not respond to the concern: if the covariance is defined in relative terms wrt. the fluxes (be it 50 or 100%), the spatial pattern is imposed because small fluxes will correspond to small variances. In other words, while calling the inversion "high resolution", the choice of the prior covariance prevents reshuffling of fluxes in the spatial domain. While this is a common choice, the paper needs to address this aspect (and point the reader to it) when discussing the sub-regional emission patterns (which are essentially just scalings of the imposed spatial prior patterns). One consequence is that flux areas that are not in the prior cannot be detected.

To address the reviewer's concern, we now add an additional sensitivity inversion, in which the prior error for emissions is specified as 50% of prior emissions or $1\times10^{-10}$ kg m$^{-2}$ s$^{-1}$ whichever is larger (as a reference a methane flux of $1\times10^{-10}$ kg m$^{-2}$ s$^{-1}$ ranks roughly 30$^{th}$ percentile in the prior inventory). This configuration allows the inversion to adjust more freely at locations with small or even no prior emissions. It is however more susceptible to noises in observations. Nevertheless, as discussed above, we find that our main findings of regional emissions are unchanged with this sensitivity test (Figure S7c and Figure S8).

(4) with constraining the total information content for the whole region to be roughly the same for the two datasets:
I find my concern still valid that, by design of the inverse method, the GOSAT and TROPOMI inversions will deliver roughly the same degrees of freedom for the entire domain (DOFs 70 and 71, Fig. 7). This is because the regularization parameter is determined by making the cost of the data and prior terms equal. While any operator is free to design such an inversion scheme, the resulting DOFs are not indicative of the actual information content of the datasets. Conclusions such as

L407: "However, over the entire East Asia domain, TROPOMI and GOSAT achieves almost the same (70) DOFS, with similar spatial patterns of averaging kernel sensitivities (Figure 7). Although the number of TROPOMI observations is much larger, strong error correlations in densely distributed data reduce the efficacy of individual observations, as shown by the difference in the regularization parameter determined for TROPOMI ($\gamma$=0.09) and GOSAT ($\gamma$=0.6) observations."

are wrong since the total DOF is just limited by the selection method of the regularization parameter. I might be mistaken here, but the authors' reply did not provide any explanations.

To address the issue raised by the reviewer, we now develop a new method to specify the $\mathbf{S_0}$ matrix, which does not require the $\gamma$ parameter which is somewhat ambiguous and difficult to interpret. We now add Section 3.3. to describe the new method and an appendix to present the mathematical derivation. In this new method, we first fully specify non-diagonal elements of $\mathbf{S_0}$ by accounting for spatial and temporal correlations, and then find an approximation to $\mathbf{S_0^{-1}}$ that is computationally tractable (direct inverse of $\mathbf{S_0}$ is difficult to compute because of its large dimension). We now use this new method in the main inversion, which should provide better interpretability than the traditional $\gamma$ method. The new method finds a higher total DOF from the TROPOMI inversion (74) than the GOSAT inversion (46). We also include a sensitivity inversion that use the old $\gamma$ method for comparison (Figure S7d and S8).

The DOFS analysis is mainly used to explain the discrepancy found in the northern India between the GOSAT and TROPOMI inversions. We now add more details to this discussion (Section 4.3.2) by showing that differences in data coverage over northern India affects how the inversion attributes downwind methane column mismatches.

(5) Other comments
   a) I am still puzzled by the small error bars e.g. on the order of 2-3% for the IND region (L234). The manuscripts states L235: "errors reported for regional estimates are 1-sigma standard deviations derived from posterior error covariance matrices". I assume that equation (4) is used, i.e. the posterior error covariance matrix comprised of smoothing error and data error. The native dimensions of the posterior covariance is 604 x 604, since the inversion is run on 600 spatial clusters plus 4 boundary elements. The discussion, however, reports fluxes and error bars for aggregated regions. How is the aggregation of the covariances carried out? Are the covariances (off-diagonal elements) duely taken into account when aggregating?

   Yes, off-diagonal elements are taken into account. The variation of an aggregated quantity is computed as $\mathbf{w}^T \hat{\mathbf{S}} \mathbf{w}$ where $\mathbf{w}$ is the aggregation vector such that the regional aggregation is given by $\mathbf{w}^T \hat{\mathbf{x}}$. We now add a brief description of how regional results are computed in Section 3.2.

b)  For the validation study, it would be good to have a figure showing the timeseries of validation and model data in addition to the statistics in the table. Given that the inversion method imposes sub-annual flux variability, I would expect quite poor agreement which might just not jump into the eye in the statistics. The poor R2 for some of the stations might be a hint.

We add the timeseries in Figure S9 following the suggestion. Figure S9 also include results from the seasonal inversions in addition to the main inversions. This shows that seasonal inversions only improve $R^2$ slightly at most sites, indicating that these poor $R^2$ is not mainly due to seasonal variations in emissions. The improvement is relatively larger at HF where the influence of rice emissions is strong. The relevant discussion is added to Section 4.2.

---

## Author Response (AR3)

**Response to Referee #3**

We thank Julia Marshall for constructive comments. Our responses are shown below in blue.

One of the most scientifically interesting findings is the temporal dependence in the differences between the GOSAT and the TROPOMI data (compared through a model), shown in Figure 6. The fact that the largest seasonal features correlate with changes of surface albedo suggests directions for the future improvement of (TROPOMI) retrievals algorithms. I think it would be worth highlighting this finding in the abstract as well.

Thanks for the suggestion. We add a sentence in the abstract highlighting this finding, which is mainly related to Northwest China and Kazakhstan.

Like the last reviewer, I am also puzzled by how small the error bars on the posterior flux estimates are. The authors mention that these uncertainties are "optimistic", as they neglect systematic uncertainties due to the inversion setup. This is not ideal, and should be improved in the future, but does not invalidate the focus of the GOSAT-TROPOMI comparison presented here.

Thanks for the comment.

**Minor comments that need to be addressed**

L326: The WLG site show(s) a relatively low posterior what? (Also, it's not the site itself that is unable to capture variability, as this sentence implies.) How can the simulations simultaneously show "relatively good agreement" and "relatively low posterior (correlation)" at WLG? Relative to what? Is it really "day-to-day" variability that is poorly captured in the case of flask data? I guess the sampling is roughly weekly, although the bizarre x-axis in S9 makes this impossible to say for sure (see comment about the supplement below).

We now change to "*Both posterior simulations … achieve **reasonable** agreement at PDI, UUM, and WLG…*" and "*…inability to capture **sub-seasonal** variability*". We have also changed the x-axis in S9 (It's now renamed to S10).

Figure 5 is lacking a colour bar!

The figure has been updated.

L506-507: Based on the error bars, only CSC and NEC show consistent (and not just qualitatively similar) posterior emissions. This is important. Either the uncertainties in the posterior emissions estimates are grossly underestimated (likely true), or the datasets are fundamentally inconsistent with one another (also likely true).

We already discuss in the manuscript the inconsistency between the two inversions over NWD

and SXJC. For BAN and CNC, the evidence that the two inversions are inconsistent is not as strong. Note the error bar shown in Figure 3 is one standard deviation (1σ).

Figure S9: Something very strange is going on with the x-axes. It looks like the observations are spaced equally, and the ticks for the months are adjusted to fit. This happens in such a way that there are always measurements on the first of each month, which is suspicious. This is not a normal way of presenting timeseries data, and should be fixed. Are there really only 23 FTIR measurements from HF for the whole year?

We have updated Figure S10 following the suggestion about x-axes. The figure shows daily-average observations. For HF, there are in total 29 days of measurements. After filtering out days with large solar zenith angles, only 23 daily averages are used for the analysis.

**Technical/stylistic comments/typos:**

We have incorporated the following comments in the revised manuscript.

- L30: Suggested rephrasing: The methane emissions inferred from GOSAT observations are … higher than those from TROPOMI observations…
- L31: These -> The
- L38: I think the "Ganges Plain" is usually referred to as the "Indo-Gangetic Plain". I would recommend changing it throughout.
- L67: a number is missing regarding the TROPOMI footprint.
- L94: Is it already clear before this study that the two sets of satellite data differ in their "regional accuracy"? If so, a citation is needed! Otherwise, remove this.
  We have changed it into "regional bias", and a citation has been added.
- L114: are -> is
- L126: annual average XCH4 on the $0.625° \times 0.5°$ grid for GOSAT and TROPOMI -> XCH4 measured by GOSAT and TROPOMI, annually averaged on the $0.625° \times 0.5°$ grid.
- L127: over THE Mongolian…
- L129: of multiple measurements fall in -> when multiple measurements fall within
- L163: located distant -> far
- L182: more discussed -> discussed further
- L183: early -> earlier (or previous)
- L204: either "a biased boundary condition" or "biased boundary conditions"
- L264: overestimate -> overestimates
- L296: estimate -> estimates
- L303: applying A traditional regularization
- L431: are in India -> either "that are in India" or "in India"
- L431: subscript in XCO2
- I would really like to see a figure of the 600 spatial clusters included in the supplement.
  See Figure S5.
- L540: benefit -> improve
- Caption of Figure S10, last line: the bias correction factor greater than -> the bias correction

factor is greater than

Brief responses to the following comments:

- L79: A bit confused by the use of "variable" here. Is it required?

  The term "variable bias" is used as in Jacob et al. (2022).

- L97, but also in abstract and throughout the study: I guess North(ern) India should be capitalized throughout? As is every region of China… This is certainly how Wikipedia does it: https://en.wikipedia.org/wiki/North_India

  We checked papers published in ACP. It appears that either northern India or North India is ok.

- L149: Were the *in situ* data temporally filtered? Generally nocturnal measurements are not well represented by models.

  Daytime measurements are used to compare with simulations. This information is now added to the text.

- L190-191: WetCHARTs is an ensemble product, which specific ensemble member was used? Is there any concern about double-counting rice emissions as wetlands?

  We use the average from the WetCHARTs ensemble. This information is now added to the text. Rice and wetland emissions used in the simulation show distinct spatial distribution. Therefore, the double-counting issue should not be a concern.

**Reference**

Jacob, D. J., Varon, D. J., Cusworth, D. H., Dennison, P. E., Frankenberg, C., Gautam, R., Guanter, L., Kelley, J., McKeever, J., Ott, L. E., Poulter, B., Qu, Z., Thorpe, A. K., Worden, J. R., and Duren, R. M.: Quantifying methane emissions from the global scale down to point sources using satellite observations of atmospheric methane, Atmos. Chem. Phys., 22, 9617-9646, https://doi.org/10.5194/acp-22-9617-2022, 2022.